# INSPECT: A Multimodal Dataset for Pulmonary Embolism Diagnosis and Prognosis

**Shih-Cheng Huang**[*]
Stanford University
mschuang@stanford.edu

**Zepeng Huo**[*]
Stanford University
zphuo@stanford.edu

**Ethan Steinberg**[*]
Stanford University
ethanid@stanford.edu

**Chia-Chun Chiang**
Mayo Clinic
chiang.chia-chun@mayo.edu

**Matthew P. Lungren**
Microsoft
mlungren@microsoft.com

**Curtis P. Langlotz**
Stanford University
langlotz@stanford.edu

**Serena Yeung**
Stanford University
syyeung@stanford.edu

**Nigam H. Shah**
Clinical Excellence Research Center
Stanford University
Technology and Digital Solutions
Stanford Healthcare
nigam@stanford.edu

**Jason A. Fries**
Stanford University
jfries@stanford.edu

## Abstract

Synthesizing information from multiple data sources plays a crucial role in the practice of modern medicine. Current applications of artificial intelligence in medicine often focus on single-modality data due to a lack of publicly available, multimodal medical datasets. To address this limitation, we introduce INSPECT, which contains de-identified longitudinal records from a large cohort of patients at risk for pulmonary embolism (PE), along with ground truth labels for multiple outcomes. INSPECT contains data from 19,402 patients, including CT images, radiology report impression sections, and structured electronic health record (EHR) data (i.e. demographics, diagnoses, procedures, vitals, and medications). Using INSPECT, we develop and release a benchmark for evaluating several baseline modeling approaches on a variety of important PE related tasks. We evaluate image-only, EHR-only, and multimodal fusion models. Trained models and the de-identified dataset are made available for non-commercial use under a data use agreement. To the best of our knowledge, INSPECT is the largest multimodal dataset integrating 3D medical imaging and EHR for reproducible methods evaluation and research[1].

## 1 Introduction

The practice of modern medicine is inherently multimodal, where synthesizing information from multiple data sources is essential for *diagnosis* (identifying which condition a patient currently has) and *prognosis* (predicting the likely course or outcome of a disease). Physicians routinely analyze patients' current symptoms and past medical history by examining imaging modalities such as X-rays and reviewing electronic health record (EHR) data to diagnose conditions, monitor disease progression, and tailor treatment plans [40, 10]. Artificial intelligence (AI) models capable of emulating the multimodal approach of contemporary medicine hold significant promise for enhancing the efficiency and accuracy of medical diagnosis, prognosis, and treatment planning.

---

[1] https://inspect.stanford.edu

37th Conference on Neural Information Processing Systems (NeurIPS 2023) Track on Datasets and Benchmarks.

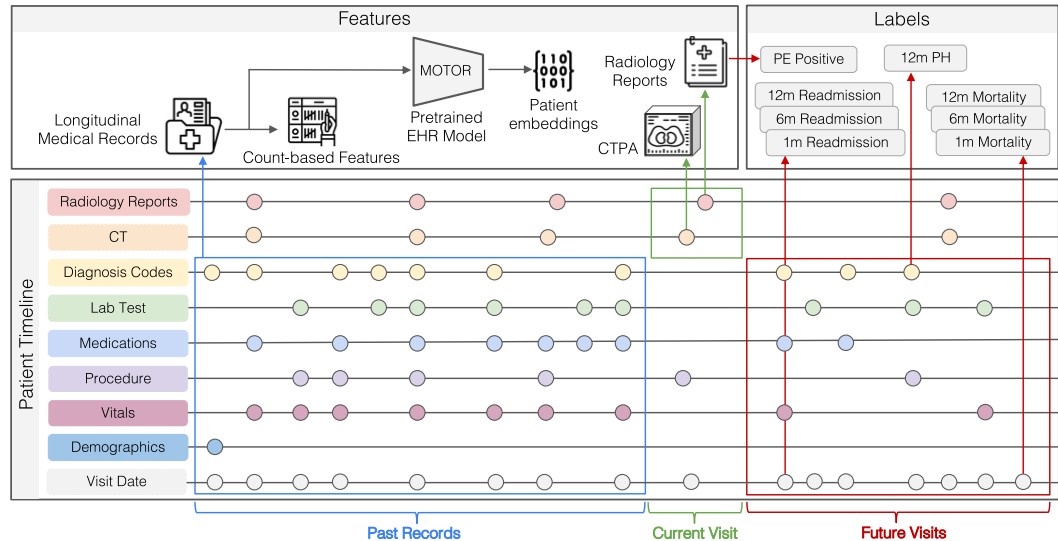

Figure 1: The INSPECT dataset comprises 19,402 patients' structured longitudinal EHRs, which includes diagnosis/procedure codes, labs, medications, vitals, and demographics, as well as 23,248 CT-scans paired with their corresponding radiology report impression section. We curated PE diagnostic and prognostic labels based on these radiology reports and subsequent visit data.

Recent advancements in *multimodal fusion* strategies have enabled medical AI models to process a wide variety of input modalities, including complimentary imaging [49, 6], EHRs [24], clinical reports [65], genomics [53], and improve model performance. Many prior works have applied multimodal fusion in medical imaging [25, 20, 14], but prognostic tasks have garnered less attention compared to diagnostic ones, primarily due to challenges in obtaining longitudinal data. For instance, prognostic tasks such as predicting 6-month mortality require future data to assign labels and involve inherently longer time horizons than diagnostic tasks. Existing medical imaging datasets are small in size [67], do not include diverse data modalities [44], or have few diagnosis/prognosis labels [38]. Addressing these limitations via new multimodal imaging datasets is essential to further advance AI-driven diagnostic and prognostic tools.

In response to these challenges, we present INSPECT (**I**ntegrating **N**umerous **S**ources for **P**rognostic **E**valuation of **C**linical **T**imelines), a multimodal dataset of patients at risk for pulmonary embolism (PE). In clinical settings, multimodal data are vital for identifying long-term complications of PE. Specifically, imaging markers from CT pulmonary angiograms (CTPA) improve prediction accuracy for adverse events [52] and combining CTPA data with EHR data increases the effectiveness of automated PE diagnosis [25]. However, the potential benefits of multimodal methods for prognosis in PE have not been fully explored, mainly due to the lack of extensive multimodal datasets with outcome labels. With INSPECT, we aim to aid in the development of new methods for multimodal fusion, tapping into both known and yet-to-be-discovered biomarkers for PE outcome prediction.

INSPECT is made available under a Data Use Agreement (DUA) for non-commercial research use. Our contributions are summarized as follows:

**1. A large-scale, multimodal medical dataset.** INSPECT contains 23,248 CTPA studies from 19,402 patients, each including: (1) high-resolution CT images (3D volumetric pixel data) with (2) paired radiology report impression sections, (3) structured longitudinal electronic health record (EHR) data and (4) clinically relevant labels, including diagnostic and prognostic labels. Each patient's longitudinal EHR provides a timeline of medical codes, demographics, medications, labs, vitals, procedures, and diagnoses. To our knowledge, this is the first dataset that combines 3D medical imaging with both radiology reports and longitudinal EHRs.

**2. A benchmark for PE diagnosis and prognosis.** We establish a benchmark for diagnosing and forecasting outcomes of pulmonary embolism through eight clinically important tasks. We assess various imaging and EHR modeling techniques, including individual models using only medical images or EHR data and combined models that use both modalities. All software and trained models used in our benchmark are available open source.

| Dataset | Modalities | | | Counts | | Curated Task Labels | |
|---|---|---|---|---|---|---|---|
| | Images | Reports | EHR | Patients | Studies | Diagnostic | Prognostic |
| UK Biobank [7] | MRI, DXA, Ultrasound | ✗ | * | 100,000 | *many* | ✗ | ✗ |
| Open-I [16] | Chest X-ray | ✓ | ✗ | 3,955 | 7,466 | ✗ | ✗ |
| CheXpert [30] | Chest X-ray | ✗ | ✗ | 65,240 | 224,316 | 14 | ✗ |
| MIMIC-CXR [32] | Chest X-ray | ✓ | ✓ | 65,379 | 227,835 | 14 | ✗ |
| RSPECT [11] | CT | ✗ | ✗ | 12,195 | 12,195 | 13 | ✗ |
| RadFusion [68] | CT | ✗ | * | 1,794 | 1,837 | 1 | ✗ |
| **INSPECT (Ours)** | CT | ✓ | ✓ | 19,402 | 23,248 | 1 | 3 |

Table 1: INSPECT vs. existing multimodal medical image datasets (* denotes partial availabilty).

## 2 Related Work

### 2.1 Medical AI for Pulmonary Embolism

Pulmonary embolism (PE) is a serious medical condition responsible for nearly 300,000 hospital admissions and approximately 180,000 fatalities each year in the United States [21]. Despite the high mortality rate associated with PE, research suggests that timely identification and commencement of appropriate treatment strategies can markedly lower both morbidity and mortality rates [41, 42]. A definitive PE diagnosis is achieved through computed tomography pulmonary angiography (CTPA) [39]; however, patients diagnosed with PE typically endure more than six days of diagnostic delay, and a quarter of these patients are misdiagnosed during their initial visit [1, 19]. Estimating long-term patient outcomes is a critical factor that can aid hospitals in efficiently allocating resources and developing an optimal patient care plan. Current practice primarily depends on rudimentary metrics such as the Pulmonary Embolism Severity Index (PESI) scoring system [54], which only considers a limited set of clinical variables.

Many studies have investigated the automation of PE detection and patient triage to alleviate the burden on radiologists [23, 25, 62, 48, 27, 58, 28]. Most of these studies do not incorporate EHR data into models, even though these records contain crucial patient history and demographic details that are essential for accurate clinical interpretation of medical images [24]. Moreover, there is a notable lack of models focused on predicting long-term outcomes for PE patients [4], largely due to the lack of publicly accessible datasets containing these labels. Hence, further research and model development that includes long-term outcome prediction and comprehensive patient EHRs have the potential to significantly improve PE detection and management.

### 2.2 Multimodal Fusion for Medical Image Applications

Incorporating clinical context is critical for accurate diagnostic interpretation of medical images. Limiting a radiologist's access to patient EHR data significantly reduces their diagnostic accuracy. Research has consistently highlighted the importance of clinical history, vitals, and lab data in accurately interpreting medical images [40, 10, 13, 12, 34]. Similarly, medical imaging models that use patient EHR data have improved accuracy and clinical relevance [24, 3, 22]. Many studies have shown the benefits of adding clinical context rather than relying solely on imaging data [64, 36, 66, 43, 26, 59, 47, 56, 29]. Nevertheless, a majority of these studies rely on a restricted subset of clinical features, primarily due to the dearth of large-scale, multimodal medical datasets. Consequently, these studies are unable to take full advantage of multimodal data fusion, highlighting the need for more comprehensive research approaches and dataset collection efforts.

### 2.3 Multimodal Datasets

Publicly available medical datasets continue to drive significant advances in medical AI research [30, 33, 68, 11]. However, very few currently available datasets include multiple modalities, are large scale, and have extensive labeled data, particularly in medical domains leveraging 3D medical images (Table 1). The limitations in data availability primarily stem from the inherent challenges associated with the release of medical data. Publicly sharing patient data requires rigorous review processes to safeguard sensitive patient information from inadvertent exposure. Furthermore, the process of labeling is often a labor-intensive and expensive endeavor. The additional challenge of managing the substantial size of 3D medical data further compounds these issues.

Among previous contributions, the MIMIC dataset [33, 31, 32] stands out as a large-scale work incorporating multiple modalities, with linkages provided with 2D chest x-rays. However, MIMIC lacks 3D medical imaging. The UK Biobank contains a variety of medical imaging modalities (e.g. MRIs, ultrasounds) combined with longitudinal medical record data [8]. However, UK Biobank imaging studies are collected prospectively, creating challenges in studying specific medical conditions, and do include corresponding radiology reports [46]. w

Radfusion [68] combines EHR summary statistics with 3D CT scans. However, Radfusion is a small-scale dataset that does not include longitudinal structured EHR data (i.e., timestamped vitals, labs, procedures, diagnoses, etc.), radiology reports, and outcome labels. Lastly, the RSNA-STR PE CT (RSPECT) dataset [11] contains 12,195 CTPA studies, but provides only a single modality, a single case per patient, and does not include prognosis labels. Our study addresses these gaps by introducing a large-scale dataset extracted from 23,248 PE cases and offers multiple modalities and labels, promising to enrich future research in this space.

# 3    Cohort Definition & Dataset Composition

Our study, approved by the Stanford Institutional Review Board (Appendix A), identified 155,950 cases involving CT pulmonary angiography at Stanford Medicine (2000-2021) using the STAnford Medicine Research Data Repository (STARR) [9]. Our cohort of CTPA cases was defined through a protocol involving random sampling, data cleaning, and inclusion criteria adherence, resulting in a final cohort of 23,248 CTPA cases for 19,402 distinct patients (see Appendix B for details). For each case, we obtained the DICOM (Digital Imaging and Communications in Medicine) files for the CT scans, the corresponding radiology reports for those scans, and structured EHR data from STARR. Each of these was then processed for analysis and de-identification. We also defined canonical training, validation, and test splits that comprise 80%, 5%, and 15% of the dataset, respectively. We defined splits based on patient IDs, such that each patient only appears in one split.

| Demographics Statistics | | | | | | | | | |
|---|---|---|---|---|---|---|---|---|---|
| **Attributes** | | **All** | **Train** | | **Val** | | **Test** | | |
| | Cases | 23,248 | 18,945 | (81.5%) | 1,089 | (4.7%) | 3,214 | (13.8%) | |
| | Patients | 19,402 | 15,789 | (81.4%) | 913 | (4.7%) | 2,700 | (13.9%) | |
| **Overlapping Studies** | RSPECT [11] | 579 | 579 | (2.5%) | 0 | (0.00%) | 0 | (0.00%) | |
| | RadFusion [68] | 772 | 772 | (3.3%) | 0 | (0.00%) | 0 | (0.00%) | |
| **Gender** | Female | 10,733 | 8,695 | (55.1%) | 517 | (56.6%) | 1,521 | (56.3%) | |
| | Male | 8,666 | 7,091 | (44.9%) | 396 | (43.4%) | 1,179 | (43.7%) | |
| | Unknown | 3 | 3 | (0.00%) | 0 | (0.00%) | 0 | (0.00%) | |
| **Age** | 0-18 | 0 | 0 | (0.0%) | 0 | (0.0% ) | 0 | (0.0%) | |
| | 18-39 | 2,912 | 2,380 | (15.1%) | 143 | (15.7% ) | 389 | (14.4%) | |
| | 39-69 | 9,974 | 8,135 | (51.5%) | 465 | (50.9% ) | 1,374 | (50.9%) | |
| | 69-89 | 5,859 | 4,740 | (30.0%) | 268 | (29.4% ) | 851 | (31.5%) | |
| | >89 | 657 | 534 | (3.4%) | 37 | (4.1% ) | 86 | (3.2%) | |
| **Race** | White | 10,704 | 8,722 | (55.2%) | 502 | (55.0%) | 1,480 | (54.8%) | |
| | Asian | 2,976 | 2,378 | (15.1%) | 152 | (16.6%) | 446 | (16.5%) | |
| | Black | 1,103 | 910 | (5.8%) | 37 | (4.1%) | 156 | (5.8%) | |
| | Native | 415 | 337 | (2.1%) | 22 | (2.4%) | 56 | (2.1%) | |
| | Unknown | 4,204 | 3,442 | (21.8%) | 200 | (21.9%) | 562 | (20.8%) | |
| **Ethnicity** | Not Hispanic | 15,628 | 12,709 | (80.5%) | 729 | (79.8%) | 2,190 | (81.1%) | |
| | Hispanic | 3,018 | 2,448 | (15.5%) | 158 | (17.3%) | 412 | (15.3%) | |
| | Unknown | 756 | 632 | (4.0%) | 26 | (2.8%) | 98 | (3.6%) | |

Table 2: Demographics statistics of the INSPECT dataset. Demographic percentages are marked in light blue. The prevalence of overlapping cases between RSPECT and Radfusion with INSPECT is also indicated. No training data from RSPECT and Radfusion are included in our validation/test sets.

| Table Type | Whole Cohort | | Per Patient | | |
|---|---|---|---|---|---|
| | # Records | Percentage | Median | Min | Max |
| Measurement | 183,820,762 | (81.5 %) | 3,783 | 0 | 500,368 |
| Drug exposure | 17,288,279 | (7.67 %) | 271 | 0 | 118,228 |
| Procedure occurrence | 8,614,273 | (3.82 %) | 190 | 1 | 35,926 |
| Condition occurrence | 8,320,211 | (3.69 %) | 148 | 0 | 27,480 |
| Visit occurrence | 5,865,211 | (2.60 %) | 126 | 1 | 16,336 |
| Visit detail | 1,355,691 | (0.60 %) | 23 | 0 | 4,840 |
| Device exposure | 88,010 | (0.03 %) | 1 | 0 | 682 |
| Person | 87,158 | (0.03 %) | 4 | 1 | 48 |
| Death | 4,410 | (0.001 %) | 0 | 0 | 13 |
| Total | 225,444,005 | (100 %) | 5,080 | 7 | 741,873 |

Table 3: Summary statistics of the longitudinal EHR data included in INSPECT.

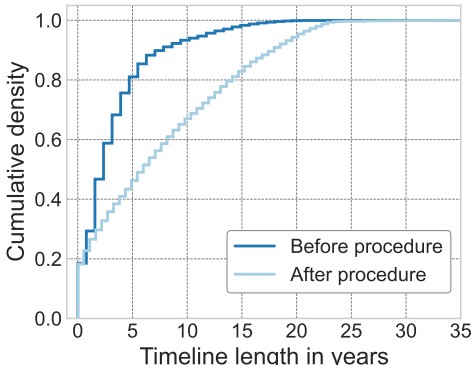

Figure 2: The cumulative probability distribution of EHR timeline lengths before and after CTPA.

Summary statistics of the demographic characteristics of our final cohort are available in Table 2. A small subset of the studies in our dataset has already been published as part of the RSPECT [11] and Radfusion [68] datasets. To maintain the integrity of our model testing process, we have ensured that no training data from the RSPECT and Radfusion datasets are included in our validation and test set. Similarly, we have also ensured that none of the validation and test data from the RSPECT and Radfusion datasets appear in our training set. This guarantees a clear separation between training and testing data, essential for unbiased model evaluation. We have indicated the prevalence of overlapping cases in Table 2. Based on this cohort, we release the following as the INSPECT dataset:

- **CTPA**: the imaging slices for the CTPAs in our cohort in DICOM format.

- **DICOM Headers**: a subset of the DICOM headers from the original DICOM file, including Patient ID, study date, instance order in the series, patient position, pixel spacing, rescale slope, rescale intercept, imaging machine manufacturer, and slice thickness. We made sure that patient ID and study date were anonymized to ensure patient privacy.

- **Radiology Report Impressions**: the impression section of the corresponding radiologist report for all the CTPAs in our cohort.

- **Structured Data From EHRs**: de-identified structured data from longitudinal EHR records for each patient in our cohort, including diagnoses, procedures, lab results, medications, and demographics. Each data element consists of a timestamp for when the event occurred, a code signifying the type of event, and optionally, a value (for lab results and vitals). All clinical events and known encounters with Stanford Health Care are included. A distribution showing the event frequency is in Table 3.

A detailed description of the formatting, hosting, and licensing details of INSPECT is in AppendixC.

# 4 Benchmark

In addition to the INSPECT dataset, we developed a benchmark for evaluating predictive models on our cohort. The code for this benchmark is included in the supplement under an open-source license.

## 4.1 Data Processing

**CTPA** Each CTPA exam is preprocessed by extracting the pixel data from the original DICOM format. We linearly transform the extracted pixel data in Hounsfield Units (HU) using the rescale slope and intercept recorded in the DICOM file. Specifically, each image $x$ is processed with $x = x * s + b$, where $s$ is the rescale slope and $b$ is the rescale intercept.

**Radiologist Reports** All CTPA exams are accompanied by a radiology report that contains a summary of the patient's medical history and detailed descriptions of the medical conditions observed by the radiologist. Using a rule-based system, we process these reports by extracting the impression section - a summary of the most important findings and possible causes. In addition, We deidentify the impression section by replacing dates and names with deidentified keywords.

**Electronic Health Records** Our source EHR records are stored in the OMOP schema [57]. They contain longitudinal EHR for patients seen at Stanford Health Care (comprised of an adult hospital and outpatient clinics) and Stanford Children's Hospital. Each record contains demographic information (age, sex, ethnicity), and coded clinical information (diagnosis codes, lab test orders and results, medication orders, procedures, and visits). The cumulative probability distribution of patient EHR timeline lengths before and after CTPA procedures is shown in Figure 2. We processed the EHR records using the FEMR (`https://github.com/som-shahlab/femr`) software package and exported data for release in FEMR's CSV format. In order to enable release, we anonymize INSPECT by introducing random time shifts for every patient, removing structured patient identifiers, and removing all unstructured text.

| O | T | L | All | Train | | Val | | Test | |
|---|---|---|---|---|---|---|---|---|---|
| **PE** | N/A | pos. | 4,689 | 3,924 | (20.7 %) | 188 | (17.3 %) | 577 | (18.0 %) |
| | | neg. | 18,559 | 15,021 | (79.3 %) | 901 | (82.7 %) | 2,637 | (82.0 %) |
| **Mort** | 1 m | pos. | 1,200 | 986 | (5.2 %) | 54 | (5.0 %) | 160 | (5.0 %) |
| | | neg. | 20,803 | 16,930 | (89.4 %) | 991 | (91.0 %) | 2,882 | (89.7 %) |
| | | cen. | 1,245 | 1,029 | (5.4 %) | 44 | (4.0 %) | 172 | (5.4 %) |
| | 6 m | pos. | 2,389 | 1,963 | (10.4 %) | 103 | (9.5 %) | 323 | (10.0 %) |
| | | neg. | 18,552 | 15,075 | (79.6 %) | 900 | (82.6 %) | 2,577 | (80.2 %) |
| | | cen. | 2,307 | 1,907 | (10.1 %) | 86 | (7.9 %) | 314 | (9.8 %) |
| | 12 m | pos. | 2,916 | 2,390 | (12.6 %) | 129 | (11.8 %) | 397 | (12.4 %) |
| | | neg. | 17,157 | 13,936 | (73.6 %) | 829 | (76.1 %) | 2,392 | (74.4 %) |
| | | cen. | 3,175 | 2,619 | (13.8 %) | 131 | (12.0 %) | 425 | (13.2 %) |
| **Re-ad** | 1 m | pos. | 857 | 695 | (3.7 %) | 38 | (3.5 %) | 124 | (3.9 %) |
| | | neg. | 20,774 | 16,898 | (89.2 %) | 997 | (91.6 %) | 2,879 | (89.6 %) |
| | | cen. | 1,617 | 1,352 | (7.1 %) | 54 | (5.0 %) | 211 | (6.6 %) |
| | 6 m | pos. | 2,185 | 1,778 | (9.4 %) | 99 | (9.1 %) | 308 | (9.6 %) |
| | | neg. | 17,953 | 14,585 | (77.0 %) | 878 | (80.6 %) | 2,490 | (77.5 %) |
| | | cen. | 3,110 | 2,582 | (13.6 %) | 112 | (10.3 %) | 416 | (12.9 %) |
| | 12 m | pos. | 2,826 | 2,291 | (12.1 %) | 130 | (11.9 %) | 405 | (12.6 %) |
| | | neg. | 16,253 | 13,201 | (69.7 %) | 794 | (72.9 %) | 2,258 | (70.3 %) |
| | | cen. | 4,169 | 3,453 | (18.2 %) | 165 | (15.2 %) | 551 | (17.1 %) |
| **PH** | 12 m | pos. | 2,726 | 2,242 | (11.8 %) | 124 | (11.4 %) | 360 | (11.2 %) |
| | | neg. | 16,503 | 13,389 | (70.7 %) | 804 | (73.8 %) | 2,310 | (71.9 %) |
| | | cen. | 4,019 | 3,314 | (17.5 %) | 161 | (14.8 %) | 544 | (16.9 %) |

Table 4: Task statistics for INSPECT. **O**: Outcome. **T**: Time Horizon. **L**: Label Value, **PE**: Pulmonary Embolism, **Mort**: In-Hospital Mortality, **Re-ad**: Re-admission, **PH**: Pulmonary Hypertension.

## 4.2 Task Definitions

Formally, given a set of multi-variate features, $\mathbf{x}_1, ..., \mathbf{x}_N$, which are encoded from the occurrences of the selected covariates in Table 3, which is a composite set of clinical events happening in continuous time steps $t_1, ..., t_N$. The goal is to train a model to approach the posterior probability of predicting the future event $t_{i+m}$ at a specific time horizon: $p[(t_{i+m}, y)|(t_i, \mathbf{x}_i)]$, where $\mathbf{x}_i$ is an accumulative feature at time $t_i$ that encodes information from features from all previous time steps $\mathbf{x}_i = f(\mathbf{x}_1, ..., \mathbf{x}_{i-1})$ and $f(\cdot)$ is the EHR modality model to be learned. The time horizon $m$ was a set of predefined periods in the future, and $y$ was a binary variable to indicate the patient having the medical event at the timestamp $t_{i+m}$. When combined with other modalities, e.g. imaging modality features, we augmented the output of EHR model $y_i^{\text{EHR}}$ at prediction time $t_i$ with a late fusion:

$$\hat{y} := \mathbf{w} \begin{bmatrix} y^{\text{EHR}} \\ \vdots \\ y^{\text{image}} \end{bmatrix}, \tag{1}$$

where the fusion weights $\mathbf{w}$ were learned from the validation set.

In the following sections, we elaborate on the precise definition for each task and how the corresponding labels are generated. Table 4 contains various statistics on the labels for each task.

### 4.2.1 Diagnostic Tasks

**Pulmonary Embolism (PE)**: We construct a pulmonary embolism diagnostic task that classifies whether pulmonary embolism is diagnosed based on the patient's CT scan. Labels are generated by applying an NLP model to the *impression* section of the corresponding radiology report. Specifically, we first fine-tune a Clinical Longformer [45] model to predict pulmonary embolism diagnoses given the impression section of a text radiology report. Ground truth labels for the reports are manually collected by [5]. Subsequently, we apply the fine-tuned model to all the impression sections of all studies in our dataset to assign a pulmonary embolism diagnosis (or lack of diagnosis) to every patient in our cohort. Appendix D describes how this model was trained and how we validated its performance.

### 4.2.2 Prognostic Tasks

For the prognostic tasks, we attempt to predict whether or not a specific event will occur in the future within a specified time horizon for a given patient. To handle missing data, patients without a recorded prognosis event and those lacking data up to the time horizon are considered censored. These patients are excluded from both training and evaluation.

The event definitions are as follows:

- **Pulmonary Hypertension (PH)**: A set of 29 International Classification of Diseases (ICD) codes to identify pulmonary hypertension.

- **In-Hospital Mortality**: We use the in-hospital mortality events provided by STARR-OMOP.

- **Re-admission**: We use the inpatient readmission events provided by STARR-OMOP.

Appendix D contains the set of ICD codes used for PH definition and outlines the methodology employed to validate this set of codes.

### 4.3 Baseline Models

We set up several common modeling approaches for each data type to serve as baselines, including image-only, EHR-only, and multimodal fusion models. The following are brief descriptions of each overall approach. The baseline model construction is shown in Figure 3. Full details, including hyperparameter tuning, can be found in Appendix F.

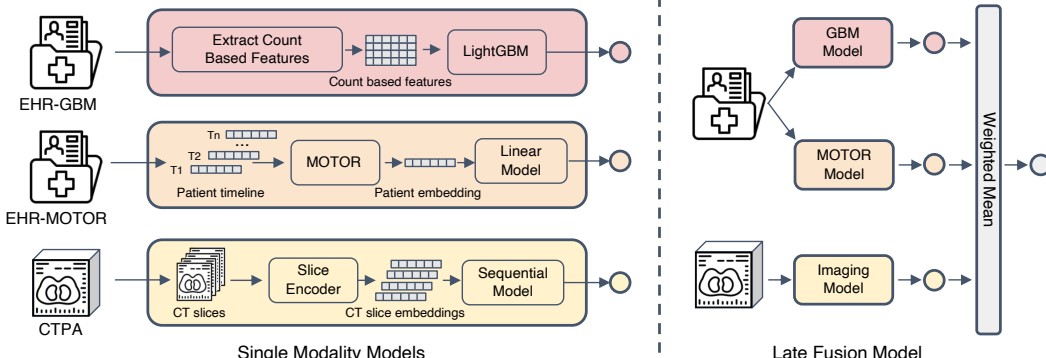

Figure 3: **Baseline Models**. We evaluate both single modality models and multi-modal late fusion models that incorporate data from both images and EHRs as baselines. For CT input, we use an LRCN (Long-term Recurrent Convolutional) model, while for structured EHR input, we employ MOTOR and gradient-boosted trees. Our multi-modal fusion baseline utilizes a late fusion approach, learning a weighted mean from each individual modality's predicted probability.

### 4.3.1 Imaging

We resize all CTPA exams to 256 by 256 pixels. Following this, we refine the focus of each slice through center cropping, resulting in a 224 by 224 pixel matrix. Subsequently, we introduce three viewing windows to highlight particular structures. Each of these windows offers an optimal view for a specific medical perspective: the lung, pulmonary embolism, and mediastinum. We then stack the three view windows into three channels, giving us an array of 224x224x3 for each CT slice. Finally, we normalize each CTPA exam to be zero-centered using ImageNet mean and standard deviation.

Figure 3 illustrates the two-step process of our CTPA model, encompassing feature extraction and sequential modeling. First, we employ a pretrained image encoder for feature extraction, processing each CT slice into a latent representation. Subsequently, all extracted features from a given CT series are inputted into a sequence encoder for our final prediction. As a baseline, we leverage a ResNetV2 model pretrained on BigTransfer [37] and finetuned on the RSPECT dataset [11] as slice encoder. For a sequence encoder, we use either an LSTM, GRU, or Transformer model [17].

### 4.3.2 Structured EHRs

**Gradient-boosted Trees**

In order to build a gradient-boosted tree model, we first featurize our structured EHR data using a common count-featurization approach [60] that creates a matrix that counts the number of times each code appears in the EHR before the index date. We then train LightGBM [35] models on these count matrix features.

**Structured EHR Foundation Model**

To address the difficulty of training a deep learning model on small datasets, we adapt MOTOR [63], a foundation model that was pretrained on Stanford structured EHR data. MOTOR was pretrained using time-to-event tasks, making it well-aligned with our desired prognostic prediction tasks. We ensure that none of the patients in our validation or test set were used for pretraining. We then fit a linear probe (i.e., a linear layer and frozen MOTOR backbone) for all tasks.

### 4.3.3 Multimodal Fusion

We evaluate multimodal fusion strategies[24] to combine our three baseline models. Our fusion models (see Figure 3) aggregate prediction probabilities from individual models by taking a weighted mean of these probabilities of each single-modality model. The weights are trained by learning a logistic regression model on the validation set using individual model probabilities as features.

| Input Modality | | | Diagnostic | Prognostic | | | | | | |
|---|---|---|---|---|---|---|---|---|---|---|
| Image | EHR | | PE | In-Hospital Mortality | | | Re-admission | | | PH |
| CT | M | G | (+) | 1 m | 6 m | 12 m | 1 m | 6 m | 12 m | 12 m |
| ✓ | | | 0.721 | 0.794 | 0.755 | 0.748 | 0.549 | 0.515 | 0.525 | 0.661 |
| | ✓ | | 0.677 | 0.923 | 0.901 | 0.892 | 0.773 | 0.779 | 0.767 | 0.824 |
| | | ✓ | 0.681 | 0.848 | 0.865 | 0.855 | 0.737 | 0.740 | 0.728 | 0.828 |
| ✓ | ✓ | | 0.761 | **0.924** | 0.903 | 0.895 | 0.774 | 0.777 | 0.764 | 0.820 |
| ✓ | | ✓ | 0.765 | 0.867 | 0.875 | 0.866 | 0.740 | 0.736 | 0.722 | 0.830 |
| | ✓ | ✓ | 0.699 | 0.922 | 0.903 | 0.892 | 0.782 | **0.786** | **0.774** | **0.849** |
| ✓ | ✓ | ✓ | **0.771** | **0.924** | **0.904** | **0.895** | **0.782** | 0.784 | 0.771 | 0.843 |

Table 5: The performance in AUROC for our different baseline modeling strategies, including various late fusions. **CT** is the CTPA based LRCN (Long-term Recurrent Convolutional Networks) model, **M** is the structured EHR based MOTOR model, and **G** is the structured EHR based gradient-boosted trees model. The best overall models are **bolded** and the best individual models are underlined.

## 5   Experiments And Results

To validate our dataset and provide some baselines, we perform experiments applying each of our three modeling strategies to each of our tasks. We also perform model fusion on each combination of individual models. Details on the computational resources used to run our experiments are in Appendix G. We also release the trained model weights learned in our experiments as part of our dataset, so our experiments can be reproduced (see Appendix C).

Table 5 contains the model performance in terms of AUROC for each of our approaches on each task. Confidence intervals can be found in Appendix H. When considering individual models, we find that the structured EHR models perform better on prognostic tasks while the CT model performs better on the diagnostic PE task. This is to be expected given that our source PE diagnoses are defined using CT scans, so the CT modality contains the information that most directly solves the diagnostic task.

We find that model fusion between the CT model and the structured EHR models helps improve performance on the diagnostic PE task but does not improve performance on the prognostic tasks. While prior work has identified imaging biomarkers related to chronic disease using CT chest imaging [51, 18], it is unclear if current deep learning models are able to take advantage of these signals. We also note that our image-only model does not match some state-of-the-art models' performance [50] on predicting PE on a similar dataset, i.e. RSPECT [11]. We posit the difference might come from nuanced label definition, where our PE definition (shown in Appendix D) has incorporated subsegmental PE where the RSPECT dataset did not.

## 6   Discussion

Our work to develop INSPECT represents a significant step forward in multimodal, multi-label, multi-timeframe medical AI research, contributing a rich, large-scale dataset and benchmarks in the context of pulmonary embolism. INSPECT encompasses diverse modalities – high-quality CT images, radiology reports, and structured EHRs – enabling the development of benchmark predictive models for PE diagnosis and prognostication. To the best of our knowledge, we are the first to link longitudinal EHR data to 3D medical images and their paired radiology reports. This dataset provides opportunities for the community to derive additional diagnostic/prognostic labels (e.g. diagnosis codes, procedure codes, lab results, medications) for either model pretraining or downstream tasks.

Our preliminary findings suggest that the integration of medical imaging and structured EHRs improves performance in diagnosing PE. However, we also find that incorporating medical imaging for prognostic tasks does not improve predictive performance, especially on the important pulmonary hypertension prediction task. These results conflict with domain knowledge and medical literature, where it has been demonstrated that CT images contain information on some of the causes of pulmonary hypertension [2]. The lack of improved performance in our study suggests that there is untapped potential in our existing techniques, either in the fundamental imaging models or the

synthesis of the imaging model output with models trained using EHR data. By releasing INSPECT , we hope to enable the research community to explore this challenge.

## 6.1 Societal Implications

The process of releasing comprehensive patient timelines carries the inherent risk of exposing identifiable information. To mitigate this risk, we have adhered to the best practices for data anonymization in accordance with HIPAA compliance standards. Further, we have released all datasets and model weights under DUA, which is a standard procedure for medical data, thus ensuring controlled access to the data and models. However, it is important to note that using these data and/or model weights to provide medical advice and or make care decisions is beyond the intended scope of use for the INSPECTdataset and associated models. To emphasize this, we have specified in our DUA that INSPECT data and models are for research purposes only, not for clinical decision-making.

## 6.2 Limitations

Firstly, INSPECT only contains data from a single site (Stanford Health Care), and models trained on INSPECT may not generalize to other patient populations. Secondly, labels are assigned based on NLP output and source EHR data, not manual chart review, and thus may be inaccurate in some cases. However, we have taken several steps to mitigate this, as detailed in Appendix D. Finally, for each CTPA image, we release only the impression section of the corresponding radiology report as de-identification protocols preclude releasing the entire note. This limits some analyses and experiments that would require entire radiology reports, beyond the impression section.

# 7 Conclusion

There are two main contributions to this work. First, we present a large-scale medical dataset INSPECT with multiple modalities, comprising health records from 19,402 patients, complete with high-quality CT images, portions of accompanying radiology reports, and structured data from patient EHRs. Second, we use this dataset to create a benchmark for a variety of important pulmonary embolism related tasks, with included baseline models. In conclusion, this work has laid the foundation for future research into multimodal fusion strategies for integrating 3D medical imaging data and patient EHR data. By openly sharing INSPECT, we hope to ignite new advances in this critical area of healthcare.

## Acknowledgments and Disclosure of Funding

Research reported in this publication was supported by the National Heart, Lung, And Blood Institute of the National Institutes of Health Awards R01HL155410 and R01HL144555, the National Library Of Medicine of the National Institutes of Health Award R01LM012966, and AIMI-HAI seed grant at Stanford University. We would also like to thank the Clinical Excellence Research Center (CERC) at Stanford for their support. The content is solely the responsibility of the authors and does not necessarily represent the official views of the National Institutes of Health.

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

# A   IRB Approval and Data De-identification

Release of INSPECT was approved by the Stanford University Institutional Review Board (IRB), given data privacy review via a standardized workflow conducted by the Center for Artificial Intelligence in Medicine and Imaging (AIMI) and the University Privacy Office. Our study was approved by the Stanford University Administrative Panel on Human Subjects Research, protocol #24883, and included a waiver of consent. All included patients from SHC signed a privacy notice, which informs them that their records may be used for research purposes given approval by the IRB.

All INSPECT data (CT scans, DICOM metadata, radiology impression sections, EHR timelines) are manually reviewed by AIMI to confirm any protected health information (PHI) is removed before public release. We de-identify each modality as follows:

**EHR Timelines:**   All dates are anonymized by using per-patient time jittering. We apply the same date transformation procedure used by MIMIC-III, specifically: "*[d]ates were shifted into the future by a random offset for each individual patient in a consistent manner to preserve intervals, resulting in stays which occur sometime between the years 2100 and 2200*" [33]. We remove all patients >89 years of age. We further remove all unstructed text fields that do not map to controlled vocabularies (e.g., SNOMED, LOINC) to prevent PHI leakage We use OHDSI Athena [61] ontologies to describe our data, which includes both public ontologies like ICD-10 as well as OHDSI specific ontologies such as Race/Gender. The full list of ontologies used is in Table 6..

| Ontology |
| --- |
| OMOP Extension |
| Medicare Specialty |
| CPT4 |
| CVX |
| ICD9Proc |
| RxNorm |
| SNOMED |
| RxNorm Extension |
| Cancer Modifier |
| ICD10PCS |
| CMS Place of Service |
| Visit |
| Ethnicity |
| Gender |
| ICDO3 |
| Race |
| LOINC |
| HCPCS |

Table 6: OHDSI Athena ontolgies used in our benchmark

**CT Scans:**   Each CT scan slice is manually reviewed for PHI, with slices containing patient information removed from the CT scan.

**Radiology Notes:**   Radiology notes are preprocessed to include only the impression section, i.e., the description of radiologist findings in the coresponding CT scan. Each note is processsed to tag names, locations, dates, telephone numbers and other HIPAA protected idenifiers such as MRNs and accession numbers. These tags are then replaced wth anonymized placeholder text. All deidentified notes are then manually reviewed to remove any additonal PHI.

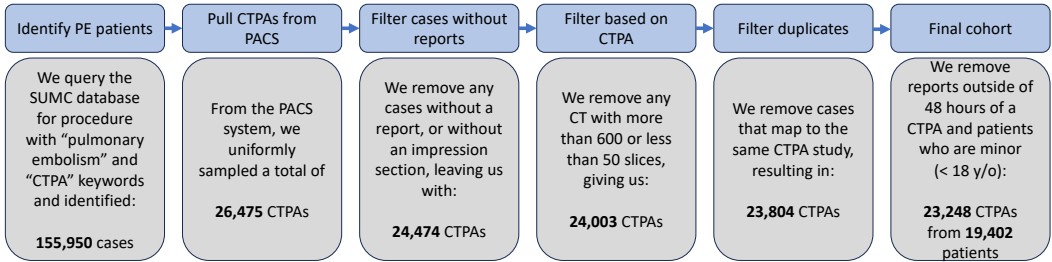

Figure 4: A flowchart of our cohort definition process.

# B  Cohort Definition

The flowchart of our cohort definition protocol is illustrated in Figure 4. With the approval from Stanford Institutional Review Board's (IRB), we identified 155,590 cases with the CT pulmonary angiography (CTPA) procedure code from the STAnford medicine Research data Repository (STARR) [15]. STARR data contains routinely collected EHR data from Stanford Health Care covering the time period of 2000 to 2021. We mapped each of these studies to their respective CTPA based on the procedure date. In instances where there was no exact match (1,296 cases) we extended the search to 10 days post-procedure date. From the mappable cases, we sampled uniformly at random 26,475 CT scans (chosen due to file storage constraints) and their corresponding radiology report.

The data cleaning phase followed, where we removed cases without a report or an impression section. This refinement process resulted in 24,474 cases for further analysis. We then selected the most relevant CTPA series per study by enforcing a slice thickness constraint between 1.0mm and 3.0mm, favoring thicker slice series. Additionally, CTs with over 600 or under 50 slices were removed from consideration. This filtering resulted in 24,003 studies. We further eliminated the test and validation data from RSNA and Radfusion in our training split and the training data thereof in our test and validation split. The patients who are minors (age < 18 years old) are removed due to privacy policy. The final selection phase involved eliminating cases with corrupted DICOMs and cases from the same patient that mapped to an identical study. The final result is a collection of 23,248 CTPA studies from a set of 19,402 unique patients.

# C  Dataset Documentation

## C.1  Hosting, Access, License, and Long-Term Preservation

We share data and trained model weights under Data Use Agreement (DUA) for non-commercial, research use. The Stanford AIMI Center will host and ensure long-term preservation of all data. Complete licensing terms for dataset and models are provided below.

INSPECT is available at https://stanfordaimi.azurewebsites.net/. We provide a preview subset of the entire dataset for reviewers. Before public release, the entire dataset is currently undergoing manual review by the AIMI Center to ensure no leakage of patient indentifying information.

As authors of the submitted dataset and corresponding manuscript, we hereby affirm that we take full responsibility for its contents. We ensure that this dataset and manuscript are original and that all data collection procedures were carried out ethically, respecting all relevant rights and regulations. We confirm that we have procured all necessary permissions for the use of the data included in the dataset, and that the data does not infringe upon any existing copyright, proprietary, or personal rights of others.

## C.2  License Terms of Use

By registering for downloads from the INSPECT Dataset, you are agreeing to this Research Use Agreement, as well as to the Terms of Use of the Stanford University School of Medicine website as posted and updated periodically at http://www.stanford.edu/site/terms/.

Permission is granted to view and use the INSPECT Dataset without charge for personal, non-commercial research purposes only. Any commercial use, sale, or other monetization is prohibited.

Other than the rights granted herein, the Stanford University School of Medicine ("School of Medicine") retains all rights, title, and interest in the INSPECT Dataset.

You may make a verbatim copy of the INSPECT Dataset for personal, non-commercial research use as permitted in this Research Use Agreement. If another user within your organization wishes to use the INSPECT Dataset, they must register as an individual user and comply with all the terms of this Research Use Agreement.

YOU MAY NOT DISTRIBUTE, PUBLISH, OR REPRODUCE A COPY of any portion or all of the INSPECT Dataset to others without specific prior written permission from the School of Medicine.

YOU MAY NOT SHARE THE DOWNLOAD LINK to the INSPECT Dataset to others. If another user within your organization wishes to use the INSPECT Dataset, they must register as an individual user and comply with all the terms of this Research Use Agreement.

You must not modify, reverse engineer, decompile, or create derivative works from the INSPECT Dataset. You must not remove or alter any copyright or other proprietary notices in the INSPECT Dataset.

The INSPECT Dataset has not been reviewed or approved by the Food and Drug Administration, and is for non-clinical, Research Use Only. In no event shall data or images generated through the use of the INSPECT Dataset be used or relied upon in the diagnosis or provision of patient care.

THE INSPECT Dataset IS PROVIDED "AS IS," AND STANFORD UNIVERSITY AND ITS COLLABORA-TORS DO NOT MAKE ANY WARRANTY, EXPRESS OR IMPLIED, INCLUDING BUT NOT LIMITED TO WARRANTIES OF MERCHANTABILITY AND FITNESS FOR A PARTICULAR PURPOSE, NOR DO THEY ASSUME ANY LIABILITY OR RESPONSIBILITY FOR THE USE OF THIS INSPECT Dataset.

You will not make any attempt to re-identify any of the individual data subjects. Re-identification of individuals is strictly prohibited. Any re-identification of any individual data subject shall be immediately reported to the School of Medicine.

Any violation of this Research Use Agreement or other impermissible use shall be grounds for immediate termination of use of this INSPECT Dataset. In the event that the School of Medicine determines that the recipient has violated this Research Use Agreement or other impermissible use has been made, the School of Medicine may direct that the undersigned data recipient immediately return all copies of the INSPECT Dataset and retain no copies thereof even if you did not cause the violation or impermissible use.

In consideration for your agreement to the terms and conditions contained here, Stanford grants you permission to view and use the INSPECT Dataset for personal, non-commercial research. You may not otherwise copy, reproduce, retransmit, distribute, publish, commercially exploit or otherwise transfer any material.

Limitation of Use: You may use INSPECT Dataset for legal purposes only.

You agree to indemnify and hold Stanford harmless from any claims, losses or damages, including legal fees, arising out of or resulting from your use of the INSPECT Dataset or your violation or role in violation of these Terms. You agree to fully cooperate in Stanford's defense against any such claims. These Terms shall be governed by and interpreted in accordance with the laws of California.

### C.3   Data Format

We detail and define our cohort in the data using a master cohort CSV file (inspect_cohort.csv), with the following primary columns.

1. patient_id: The de-identified patient id
2. procedure_time: The date of the CTPA procedure, in ISO 8601 format
3. split: A string, either "train", "valid", or "test" that indicates the data split for this patient

The primary keys for our cohort are patient_id and procedure_time. Every case, label, and feature set is associated with a patient_id / procedure_time pair.

This file also includes the following demographic columns: age, gender, race, ethnicity.

We additionally include various columns for labels.

First, we include three NLP based pulmonary embolism diagnostic label columns. pe_positive_nlp is the main PE label used in all of our experiments and described as "Positive PE" in Appendix D.1. pe_acute_nlp and pe_subsegmentalonly_nlp are the other two label columns that are similarly described in Appendix D.1. Every case in our cohort is assigned either "True" or "False" for each of these columns.

Second, we include seven prognostic label columns. These label columns correspond to the seven prognostic tasks in our paper and have the following names: 1_month_mortality, 6_month_mortality, 12_month_mortality, 1_month_readmission, 6_month_readmission, 12_month_readmission 12_month_PH. Every case in our cohort is assigned either "True", "False", or "Censored" for each of these columns.

Finally, we include indicators for whether or not each case in this dataset is also present in either of the RNSA or Radfusion datasets using the RSNA and radfusion columns respectively.

### C.3.1 CTPA

The CTPAs are made available in DICOM format. To ensure patient privacy, patient ID and study date were anonymized, and only a subset of the DICOM headers from the original DICOM file are included: ['InstanceNumber', 'ImagePositionPatient', 'PixelSpacing', 'RescaleIntercept', 'RescaleSlope', 'WindowCenter', 'WindowWidth', 'Manufacturer', 'PhotometricInterpretation', 'SliceThickness']

### C.3.2 Radiologist Report Impressions

Radiologist reports with impressions sections, after the anonymization process, are included in a CSV file with the name INSPECT_anon_impression.csv.

It contains columns with the names: PatientID, StudyTime and anon_impression, where the first two columns are used to map to the master cohort file and anon_impression is the anonymized impression section.

### C.3.3 Structured EHR Data

Structured EHR data is released as gzipped CSV files in the FEMR format, which is documented at `https://github.com/som-shahlab/femr/blob/main/tutorials/2b_Simple_ETL.ipynb`. We release all known diagnoses, procedures, lab tests, medications, visits, and death records for patients in our cohort. The FEMR format is a simplified subset of OMOP 5.3 [57], with a subset of the columns and tables. It can be processed with any CSV reader as well as with the FEMR software package.

### C.4 Dataset Statistics

### C.4.1 CTPA

Based on our inclusion criteria, each CTPA study can have between 50 to 600 slices. On average, each CTPA has 220.6 slices, giving us a total of 5,164,472 CT slices in our dataset. The CTPA studies range from 1.00mm to 3.00mm (Table 7) collected from CT scanners by 3 different manufacturers (Table 8).

| Slice Thickness | Count |
|---|---|
| 3.00 | 8,600 |
| 2.50 | 2,840 |
| 2.00 | 8 |
| 1.50 | 5,366 |
| 1.25 | 7,834 |
| 1.00 | 16,911 |

Table 7: Slice Thickness Distribution

### C.4.2 Structured EHR Data

**Distribution of history and follow-up times** Our released EHR data contains all of Stanford's records for each patient in our cohort. As such, we have relatively substantial history before and follow-up time after each CTPA procedure in our cohort. Figure 5 provides the distributions for both the amount of history and the amount of follow-up time in days for our dataset. For the patients who

| Manufacturer | Count |
|---|---|
| SIEMENS | 11,357 |
| GE MEDICAL SYSTEMS | 3,786 |
| TOSHIBA | 3,072 |

Table 8: CT Scanner Manufacturer Distribution

| | Min | Max | Average | Standard deviation |
|---|---|---|---|---|
| Interval (in days) | 0 | 6887 | 448.19 | 764.87 |

Table 9: Statistics of CTPA scans intervals across all patients

underwent multiple CTPA scans we also calculate some basic statistics of the intervals, shown in Table 9.

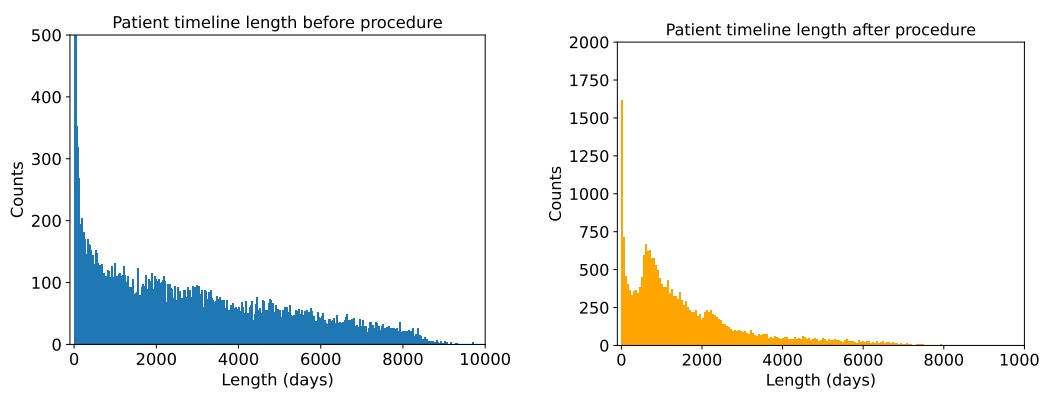

Figure 5: Patient timeline length distributions

**Distribution of data types** Here we show the types of data (clinical events) in the EHR patient timelines (Figure 6). We can see the measurement table dominates the OMOP table distribution and is larger than others by an order of magnitude.

### C.5 Model Releases

To aid reproducibility, we release all models trained in our experiments as part of the dataset.

EHR models are in the form of pickle objects, either LightGBM Classifiers for the LightGBM models or sklearn LogisticRegression for the linear probes for MOTOR.

CT models are saved in the form of PyTorch checkpoints.

## D Task Label Definitions And Validation

As part of our project, we developed and validated a set of diagnostic labels for pulmonary embolism and a set of prognostic labels for the risk of pulmonary hypertension for every case in our dataset.

### D.1 Pulmonary Embolism

**Task Label Definition**

We construct three sets of pulmonary embolism labels ("Positive PE", "Subsegmental PE", and "Acute PE"). All the primary analysis in the benchmark is done using "Positive PE", but we release all three

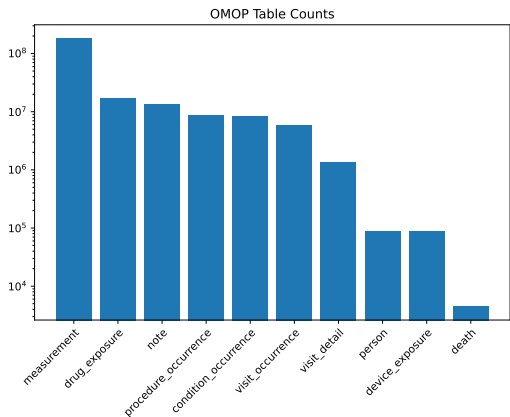

Figure 6: OMOP table distribution

as part of our dataset. We define these labels using the impression section of radiology reports as ground truth. If the radiology report contains evidence of the label, we consider that to be a positive example. If the radiology report is either unclear or contains evidence against the label, we consider that a negative. We shifted the prediction time of PE to 24 hours before the CTPA exam time to avoid feature leakage, following [30].

To develop our NLP PE labeler, we use the annotated dataset described in Banerjee et al. 2019 [5]. This dataset includes 4,351 CTPA reports obtained from Stanford Healthcare Center between 2000-2016. All reports were manually annotated by board-certified radiologists according to the following labels:

- **Positive PE**: This label is critical as it signifies the presence of a PE, a potentially life-threatening condition where one or more of the pulmonary arteries in the patient's lungs is blocked by a blood clot. Accurate identification of PE in radiology reports is a crucial step towards timely treatment and patient recovery.

- **Subsegmental PE**: This label indicates a PE that affects the subsegmental branches of the pulmonary arteries, the smaller vessels within the lung. This label is essential in tailoring patient treatment as subsegmental PE sometimes have different treatment protocols compared to PE located in the larger pulmonary arteries. The classification of PE down to the subsegmental level is vital for precision medicine.

- **Acute PE**: This label marks a sudden onset of PE. Acute PE is particularly significant due to its immediate risk to the patient. Rapid identification and treatment of acute PE can mean the difference between life and death. As such, the Acute PE label serves as an urgent signal in the patient's radiology report, prompting immediate medical intervention.

Using these reports and labels, we train a text-based labeling model that can automatically label the impression sections of radiology reports with "Positive PE", "Subsegmental PE", and "Acute PE" labels. We utilize a pretrained version of the Clinical Longformer model [45], as the backbone of our NLP labeler. This model is then finetuned, validated, and tested using our hand-labeled cases. After finetuning, this model is applied to generate labels for every case in our dataset, with the "Positive PE" label in particular used for all analyses. Each label was deemed positive if the model's prediction probability exceeded 0.5; otherwise, the label was classified as negative. The labeling process is shown in Figure 7.

**Task Label Validation**

We validate our PE NLP labels using the test set of the manual labels. The performance of the model can be found in Table 10. The precision and recall are quite high, especially for the main "Positive PE" label, indicating that our NLP-generated labels are high quality.

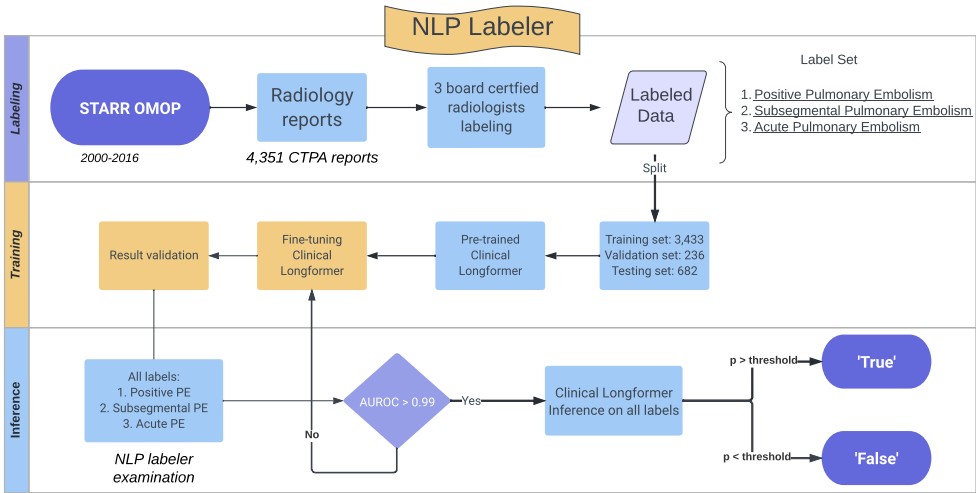

Figure 7: A flowchart of our NLP labeling process.

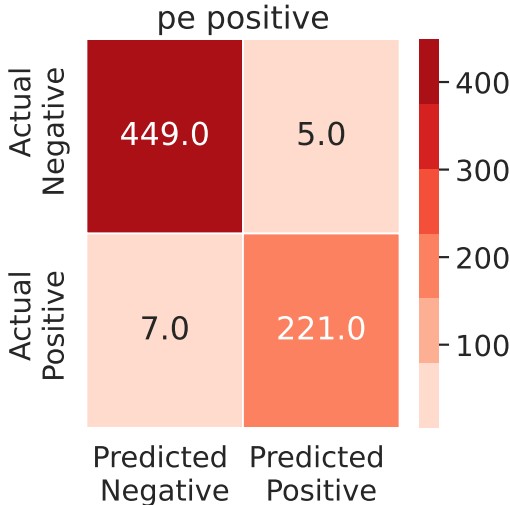

Figure 8: Confusion matrix between NLP labeler and human experts on notes under test set of our training data for NLP labeler

Even with satisfactory performance, we are interested to know the error modes of our NLP labeler so we manually examine the predictions of it against the ground truth. The confusion matrices are shown in Figure 8. When comparing against human annotation on notes, for false positive cases, we observed that the NLP might have mistakenly used the wording '...*thrombus within the superior vena cava...*' as an indicator of positive pulmonary embolism when it is not. For false negative cases, the NLP labeler might have incorrectly used the wordings '...*possible pulmonary emboli are incompletely evaluated. Consider ct pe protocol if clinically indicated...*' as positive PE. Overall the misclassifications when compared to experts' annotated notes are relatively low (12 out of 682).

## D.2 Pulmonary Hypertension

**Task Label Definition**

We construct a set of prognostic pulmonary hypertension labels that attempt to define whether or not a patient has a future incidence of pulmonary hypertension in the next year.

|  | Positive PE | Subsegmental PE | Acute PE |
|---|---|---|---|
| AUROC | 0.99 | 0.99 | 0.99 |
| F1 | 0.97 | 0.95 | 0.96 |
| Precision | 0.97 | 0.98 | 0.98 |
| Recall | 0.98 | 0.93 | 0.94 |
| Accuracy | 0.98 | 0.99 | 0.98 |
| Class counts (pos) | 228 | 43 | 201 |
| Class counts (neg) | 454 | 639 | 481 |
| Total support | 682 | 682 | 682 |

Table 10: NLP PE labeler performance. Each label was deemed positive if the model's prediction probability exceeded 0.5; otherwise, the label was classified as negative.

|  | # Patients |
|---|---|
| Positive PH | 97 |
| Negative / Unknown PH | 23 |

Table 11: Statistics for our ground truth hand-labeled pulmonary hypertension labels.

We start by having a board-certified clinician create a manually annotated label set for this task on 120 patients in our cohort. We define ground truth for this task based on the review of a subset of notes for those patients. Each of these notes is either labeled "Positive" or "Negative", where "Positive" is that the patient has pulmonary hypertension and "Negative" is that the patient either doesn't have pulmonary hypertension or it is unknown. We then aggregate these labels at the patient level, labeling a patient with any "Positive" label as "Positive" and "Negative" otherwise. The statistics for these labels are in table 11.

Hand-labeling all of the cases in our dataset is not viable so we use this seed set of manual labels to develop a structured data-based phenotyping algorithm that can then be applied to all of the patients in our dataset. From manual review and expert assistance, we derive Table 12 which contains a comprehensive list of ICD and internal Stanford codes that can be used to identify pulmonary hypertension. This phenotyping algorithm is applied to obtain the pulmonary hypertension labels that we use for our primary analysis.

**Task Label Validation**

We validate our pulmonary hypertension phenotyping algorithm by testing it using the hand-labeled set. Our hand labels don't incorporate time, so we can't directly compare the 12-month PH task used in our analysis to them. Instead, we compare a slightly modified algorithm that uses the same ICD/Stanford code list to identify patients who have ever had PH and compare that set of patients to the set of "Positive" patients in our hand-labeled set. The precision, recall, F1 and accuracy are in Table 13. Our phenotyping algorithm has a very high recall, of 0.91, with slightly worse precision. The reduced precision is likely due to how we only hand labeled a subset of notes, so our ground truth here has poor recall. Regardless, this demonstrates that the structured data phenotyping algorithm we are using is effective.

# E   Example of CTPA scan

For the readers who are unfamiliar with CTPA, we also attached an example in Figure 9. This scan demonstrates an MPR (multi-planar reconstruction) format rendering of the 3D volumetric CTPA scan from our INSPECT cohort.

Table 12: Concepts of pulmonary hyptertension and their ICD9/10 codes

| Concept Name | Vocabulary ID | Code |
|---|---|---|
| Pulmonary hypertension | STANFORD_CONDITION | 1029634 |
| Secondary pulmonary arterial hypertension | ICD10CM | I27.21 |
| Pulmonary hypertension due to left heart disease | ICD10CM | I27.22 |
| Chronic pulmonary heart disease | ICD9CM | 416 |
| Kyphoscoliotic heart disease | ICD9CM | 416.1 |
| Chronic pulmonary embolism | ICD9CM | 416.2 |
| Other secondary pulmonary hypertension | ICD10 | I27.2 |
| Other secondary pulmonary hypertension | ICD10CM | I27.29 |
| Eisenmenger's syndrome | ICD10CM | I27.83 |
| Primary pulmonary hypertension | ICD10 | I27.0 |
| Primary pulmonary hypertension | ICD9CM | 416.0 |
| Other chronic pulmonary heart diseases | ICD9CM | 416.8 |
| Other specified pulmonary heart diseases | ICD10CM | I27.89 |
| Chronic pulmonary embolism | ICD10CM | I27.82 |
| Pulmonary hypertension due to alveolar hypoventilation disorder | STANFORD_CONDITION | 1170535 |
| Kyphoscoliotic heart disease | ICD10CM | I27.1 |
| Pulmonary hypertension, unspecified | ICD10CM | I27.20 |
| Pulmonary hypertension due to lung diseases and hypoxia | ICD10CM | I27.23 |
| Chronic pulmonary heart disease, unspecified | ICD9CM | 416.9 |
| Cor pulmonale (chronic) | ICD10CM | I27.81 |
| Pulmonary heart disease, unspecified | ICD10CM | I27.9 |
| Pulmonary heart disease, unspecified | ICD10 | I27.9 |
| Primary pulmonary hypertension | ICD10CM | I27.0 |
| Kyphoscoliotic heart disease | ICD10 | I27.1 |
| Secondary pulmonary hypertension | STANFORD_CONDITION | 67294 |
| Chronic pulmonary heart disease (CMS-HCC) | STANFORD_CONDITION | 142308 |
| Chronic thromboembolic pulmonary hypertension | ICD10CM | I27.24 |
| Other secondary pulmonary hypertension | STANFORD_CONDITION | 2065632 |
| Other secondary pulmonary hypertension | ICD10CM | I27.2 |

.

|  | Positive PH |
|---|---|
| F1 | 0.88 |
| Precision | 0.85 |
| Recall | 0.91 |
| Accuracy | 0.80 |

Table 13: Structured data-based PH labeler performance.

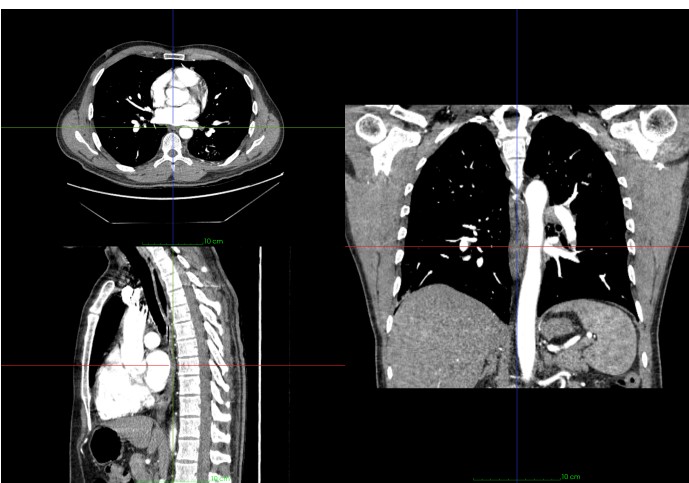

Figure 9: An example of CTPA examination scan in multi-planar reconstruction

| Hyperparameters | Values |
|---|---|
| **LightGBM** | |
| max_depth | 3, 6, -1 |
| learning_rate | 0.02, 0.1, 0.5 |
| num_leaves | 10, 25, 100 |
| software_version | LightGBM 3.3.5 |
| **MOTOR** | |
| linear_probe_l2_strength | automatic between 10 and 0 |
| dropout | 0 |
| learning_rate | $10^{-5}$ |
| num_time_bins | 8 |
| survival_dim | 512 |
| inner_dim | 768 |
| layers | 12 |
| max_sequence_length | 16,384 |
| vocabulary_size | 65,536 |
| software_version | femr 0.1.8 |
| **CTPA Slice Encoder** | |
| learning_rate | 0.0005 |
| optimizer | AdamW |
| loss | BCEWithLogitsLoss |
| architecture | resnext101_32x8d |
| pretrain data | BigTransfer |
| software_version | timm 0.9.2 |
| **CTPA Sequence Encoder** | |
| learning_rate | 0.001, 0.0005, 0.0001, 0.00005 |
| n_epochs | 50 |
| slice aggregation | max, mean, attention, attention+max |
| sequence encoder type | LSTM, GRU, Transformer |
| hidden size | 128, 256, 512 |
| bidirectional | True, False |
| num_layers | 1, 3, 5 |
| dropout_prob | 0.0, 0.25, 0.5, 0.75 |
| weighted_sampling | True, False |
| pretrain data | RSNA RESPECT |
| input_sie | 256 |
| **PE NLP Labeler** | |
| max_sequence_length | 1536 |
| learning_rate | 2e-5 |
| n_epochs | 15 |
| architecture | Longformer |
| pretrain type | Clinical-Longformer |
| software_version | hugging face 4.30.1 |

Table 14: Hyperparmeter search grids of methods under comparison in our experiments. The software version for implementing each method is also shown.

# F  Additional Model Details And Hyperparameters

Hyperparameters are selected through grid search on the validation set. Table 14 contains the hyperparameter grids, and the software versions used for each model.

## F.1  CTPA model

**Windowing** We here begin to describe the viewing window for our CTPA imaging model. (window center = -600, window width = 1500), pulmonary embolism (window center = 400, window width = 1000), and the mediastinum (window center = 40, window width = 400). Specifically, for each viewing window, we clipped the Hounsfield Unit (HU) pixel values to fall within the range $[windowcenter - windowwidth/2, windowcenter + windowwidth/2]$

**Slice Encoder Augmentations** After the windowing operation, every CT scan is resized to dimensions of 256x256 followed by a random cropping operation to yield a 224x224 size. Before inputting into the model, each slice is normalized using the mean and standard deviation values from ImageNet. Once the slice encoder training phase is concluded, each slice is inputted into the trained model for the extraction of a latent representation. In this phase, center cropping is applied as opposed to random cropping for retrieving slice representations.

**Sequence Encoder Augmentations** Before the slice representations are input into a sequence encoder, we ensure each series is standardized to the same input size through either random sampling or padding. Specifically, if a series has a higher slice count than num_slices, a random sampling of the slices is conducted to equalize with num_slices. Alternatively, if a series possesses fewer slices, padding is executed with zero vectors to complete the series.

## F.2  Structured Electronic Health Records Models

### Gradient Boosted Tree Model

For our featurization, we use count featurization augmented by ontologies. For count featurization, we count each occurrence before the prediction time of every medical code (diagnoses, procedures, lab orders, and medications) and have a column containing the count for each code. Normally, this is a very sparse matrix as each code individually is relatively rare, so we take advantage of the standard *ontology expansion* technique, where we count higher level concepts in addition to the raw codes themselves. For instance, we will have a column both for the number of very specific ICD/I27.29 codes as well as a column for the more generic ICD/IXX (and I class ICD code) concept.

These features are then fed into a hyperparmeter tuned LightGBM model [35].

### MOTOR Model

MOTOR [63] is a self-supervised transformer model designed for long-term medical prediction. For our experiments, we use a version that was already pretrained on de-identified Stanford data. We explicitly construct our training, validation, and test cohorts in sync with that pretrained model such that there is no overlap between its pretraining data and our test and validation data.

We use the linear probe method for adapting MOTOR to our tasks. Aka, we extract the final patient representation from the last transformer layer and then train a logistic regression model with L2 regularization on those representations.

## F.3  Model Fusion

For model fusion, we apply a simple late fusion strategy of taking a weighted average of the outputs of each source model. We implement this by first converting all output probabilities to logits, and then fitting a logistic regression model on those logits using the validation set. We do not use any regularization for that logistic regression model as it only has at most 3 features in our setup.

Furthermore, we examine the agreement and disagreement between the three source models by computing the Spearman correlations between their output probabilities on the 8 tasks in our dataset. Figure 10 contains the corresponding heatmaps. As expected, the two EHR based models, MOTOR and GBM, are much more correlated with each other than the CT based model.

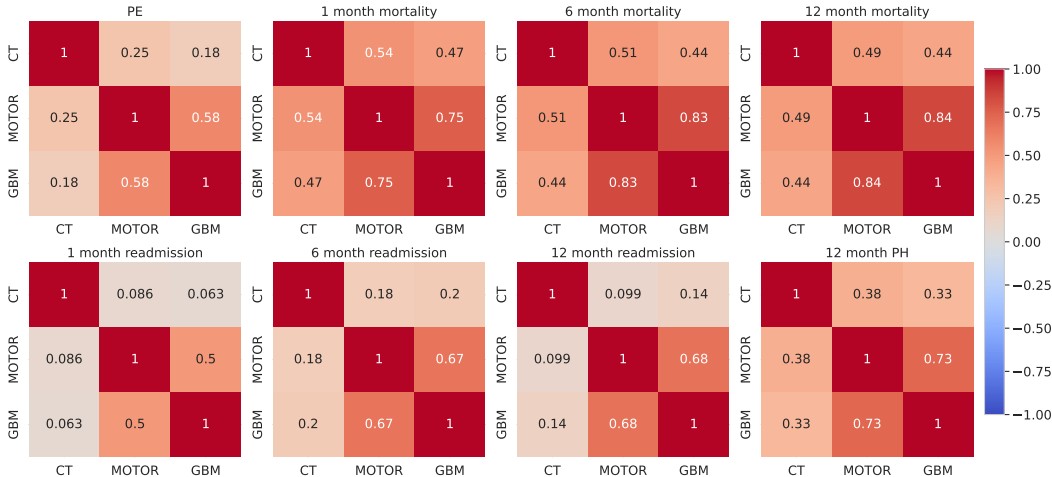

Figure 10: Spearman correlation matrices for each pair of models' output probabilities. **CT** is the CTPA based LRCN model, **M** is the structured EHR based MOTOR model, and **G** is the structured EHR based gradient-boosted trees model.

| Input Modality | | | Diagnostic | Prognostic | | | | | | | |
|---|---|---|---|---|---|---|---|---|---|---|---|
| Image | EHR | | PE | In-Hospital Mortality | | | Re-admission | | | PH |
| CT | M | G | (+) | 1 m | 6 m | 12 m | 1 m | 6 m | 12 m | 12 m |
| ✓ | | | (0.69, 0.75) | (0.76, 0.83) | (0.73, 0.78) | (0.72, 0.77) | (0.50, 0.60) | (0.48, 0.55) | (0.49, 0.56) | (0.63, 0.69) |
| | ✓ | | (0.66, 0.70) | (0.91, 0.94) | (0.89, 0.92) | (0.88, 0.91) | (0.73, 0.81) | (0.75, 0.81) | (0.74, 0.79) | (0.80, 0.85) |
| | | ✓ | (0.66, 0.71) | (0.82, 0.87) | (0.85, 0.88) | (0.84, 0.87) | (0.69, 0.78) | (0.71, 0.77) | (0.70, 0.75) | (0.80, 0.85) |
| ✓ | ✓ | | (0.74, 0.78) | (0.91, 0.94) | (0.89, 0.92) | (0.88, 0.91) | (0.73, 0.82) | (0.75, 0.80) | (0.74, 0.79) | (0.80, 0.84) |
| ✓ | | ✓ | (0.74, 0.79) | (0.84, 0.89) | (0.86, 0.89) | (0.85, 0.88) | (0.69, 0.78) | (0.71, 0.76) | (0.70, 0.75) | (0.81, 0.85) |
| | ✓ | ✓ | (0.68, 0.72) | (0.91, 0.94) | (0.89, 0.92) | (0.88, 0.91) | (0.74, 0.82) | (0.76, 0.81) | (0.75, 0.80) | (0.82, 0.87) |
| ✓ | ✓ | ✓ | (0.75, 0.79) | (0.91, 0.94) | (0.89, 0.92) | (0.88, 0.91) | (0.74, 0.82) | (0.76, 0.81) | (0.75, 0.79) | (0.82, 0.86) |

Table 15: 95% confidence intervals as a function of the test set for model performance in AUROC. **CT** is the CTPA based LRCN model, **M** is the structured EHR based MOTOR model, and **G** is the structured EHR based gradient-boosted trees model.

## G   Experiment Compute Environment

EHR experiments are performed in a local on-prem university compute environment using 24 Intel Xeon 2.70GHz CPU cores and 1 Nvidia V100 GPU.

Image experiments are performed on a HIPAA-compliant Google virtual machine using 4 x Nvidia V100 GPU with 96 Intel Skylake vCPU with 624GB of RAM.

All compute environments supported HIPAA-compliant data protocols.

## H   Confidence Intervals

We obtain some uncertainty estimates for our results by bootstrapping with respect to the test set.

First, we estimate the uncertainty of the AUROC of each model. For each task, we create 1,000 bootstrap samples and compute the AUROC for each model on each bootstrap sample. We then extract the 2.5% and 97.5% percentiles of the 1,000 samples to obtain 95% confidence intervals.

Table 15 presents the results of this analysis. The widths of the intervals are often around 0.04, indicating that we are able to estimate model performance with reasonable precision.

Second, we estimate the uncertainty of the relative AUROC of each model. We use the same bootstrap samples as in the first analysis, but compute the relative performance between each model and our chosen baseline, which we arbitrarily choose as MOTOR.

| Input Modality | | Diagnostic | Prognostic | | | | | | |
|---|---|---|---|---|---|---|---|---|---|
| Image | EHR | PE | In-Hospital Mortality | | | Re-admission | | | PH |
| CT | M G | (+) | 1 m | 6 m | 12 m | 1 m | 6 m | 12 m | 12 m |
| ✓ | | **(0.01, 0.07)** | **(-0.16, -0.10)** | **(-0.17, -0.12)** | **(-0.17, -0.12)** | **(-0.29, -0.16)** | **(-0.30, -0.22)** | **(-0.28, -0.20)** | **(-0.19, -0.13)** |
| | ✓ | (0.00, 0.00) | (0.00, 0.00) | (0.00, 0.00) | (0.00, 0.00) | (0.00, 0.00) | (0.00, 0.00) | (0.00, 0.00) | (0.00, 0.00) |
| | ✓ | (-0.02, 0.03) | **(-0.10, -0.05)** | **(-0.05, -0.02)** | **(-0.05, -0.02)** | (-0.08, 0.01) | **(-0.06, -0.02)** | **(-0.06, -0.02)** | (-0.02, 0.03) |
| ✓ | ✓ | **(0.06, 0.10)** | (-0.00, 0.01) | (-0.00, 0.01) | (-0.00, 0.01) | (-0.00, 0.00) | (-0.00, 0.00) | (-0.01, 0.00) | (-0.01, 0.00) |
| ✓ | ✓ | **(0.06, 0.11)** | **(-0.08, -0.03)** | **(-0.04, -0.01)** | **(-0.04, -0.01)** | (-0.08, 0.01) | **(-0.07, -0.02)** | **(-0.07, -0.02)** | (-0.01, 0.03) |
| | ✓ ✓ | **(0.01, 0.03)** | (-0.00, 0.00) | (-0.00, 0.00) | (-0.00, 0.00) | (-0.00, 0.02) | **(0.00, 0.01)** | **(0.00, 0.01)** | **(0.01, 0.04)** |
| ✓ | ✓ ✓ | **(0.07, 0.11)** | (-0.00, 0.01) | (-0.00, 0.01) | (-0.00, 0.01) | (-0.00, 0.02) | **(0.00, 0.01)** | (-0.00, 0.01) | **(0.00, 0.04)** |

Table 16: 95% confidence intervals for the difference in AUROC performance between a particular model and the structured EHR based MOTOR model. **CT** is the CTPA based LRCN model, **M** is the structured EHR based MOTOR model, and **G** is the structured EHR based gradient-boosted trees model. Statistically significant differences at p = 0.05 are **bolded**.

| Input Modality | | Diagnostic | Prognostic | | | | | | |
|---|---|---|---|---|---|---|---|---|---|
| Image | EHR | PE | In-Hospital Mortality | | | Re-admission | | | PH |
| CT | M G | (+) | 1 m | 6 m | 12 m | 1 m | 6 m | 12 m | 12 m |
| ✓ | | 0.715, (0.003) | 0.741, (0.007) | 0.753, (0.001) | 0.750, (0.003) | 0.549, (0.008) | 0.547, (0.014) | 0.551, (0.012) | 0.658, (0.005) |

Table 17: The mean and standard deviation in AUROC for the various tasks when the random seed is changed. We use 5 random seeds to estimate both the mean and standard deviation.

Table 16 contains these relative confidence intervals. Most (12/14) of the intervals for individual models exclude zero, indicating that we have enough precision to accurately tell the difference in performance between CT, MOTOR, and gradient-boosted tree models. The fused models have a less clear separation, with about half of the differences being statistically insignificant.

In order to conduct variation study, we have rerun our image-based modality for 5 times for different seeds. Note that our EHR baselines, MOTOR and LightGBM, are deterministic with the hyperparameters we used in our study, so we do not perform reseeding experiments. The results of this analysis are in Table 17.

## I  Model Performance as a Function of PE Status

In our results section, we present performance statistics on the entire cohort. However, it is sometimes useful to look at performance within patients who test positive for PE (+) vs patients who test negative for PE (-). Table 18 contains the performance on the seven prognostic tasks by PE status.

## J  Comparison of Models vs. Simplified PESI

Clinical risk scores are heuristics commonly used in medicine to inform treatment decisions. We compare our machine learning-based models against a common PE rule-based risk calculator, the simplified PESI (sPESI) score [55]. sPESI is a 0-6 scoring rule comprised of the following additive criteria (each rule contributes +1 to the overall score) in Table 19.

To ensure that the features used to calculate sPESI reflect the patient's condition at the time of the imaging, we only use data between the most recent 10 days prior to the CT exam and the 2 days after the CT scan for the numeric sPESI features. Patients that are missing data required for sPESI are dropped. In addition, as sPESI is only designed for use with patients that have PE, we further restrict this analysis to patients that have a positive diagnosis for PE. This results in a total of 1,719 cases derived from 1,609 unique patients with the required sPESI features and PE. The amount of patients with each total score is listed in Table 20.

We evaluate the sPESI score by measuring the performance in terms of AUROC of using the score to rank patients for our seven prognostic tasks. Table 21 contains the results of this comparison. Note that sPESI is designed for short-term mortality prediction, and might not be meaningful in the context of other prognostic tasks. We observe relatively low performance for the sPESI. One potential cause of that low performance is that our retrospective data has a much higher degree of missingness

| Has PE | Input Modality | | | Prognostic | | | | | | |
|---|---|---|---|---|---|---|---|---|---|---|
| | Image | EHR | | In-Hospital Mortality | | | Re-admission | | | PH |
| | CT | M | G | 1 m | 6 m | 12 m | 1 m | 6 m | 12 m | 12 m |
| (+) | ✓ | | | 0.761 | 0.738 | 0.726 | 0.609 | 0.586 | 0.629 | 0.596 |
| | | ✓ | | 0.914 | 0.897 | 0.869 | 0.782 | 0.770 | 0.755 | 0.761 |
| | | | ✓ | 0.853 | 0.850 | 0.835 | 0.773 | 0.763 | 0.729 | 0.762 |
| | ✓ | ✓ | | 0.879 | **0.902** | 0.870 | 0.752 | 0.766 | **0.760** | 0.740 |
| | ✓ | | ✓ | 0.817 | 0.858 | 0.844 | 0.712 | 0.748 | 0.732 | 0.740 |
| | | ✓ | ✓ | **0.914** | 0.897 | **0.871** | **0.788** | **0.789** | 0.757 | **0.772** |
| | ✓ | ✓ | ✓ | 0.879 | 0.899 | 0.870 | 0.762 | 0.776 | 0.758 | 0.752 |
| (-) | ✓ | | | 0.806 | 0.758 | 0.754 | 0.534 | 0.504 | 0.507 | 0.677 |
| | | ✓ | | 0.925 | 0.902 | 0.897 | 0.771 | 0.782 | 0.770 | 0.852 |
| | | | ✓ | 0.847 | 0.869 | 0.861 | 0.728 | 0.737 | 0.728 | 0.852 |
| | ✓ | ✓ | | **0.927** | 0.903 | 0.900 | 0.771 | 0.778 | 0.764 | 0.850 |
| | ✓ | | ✓ | 0.872 | 0.879 | 0.873 | 0.729 | 0.728 | 0.716 | 0.859 |
| | | ✓ | ✓ | 0.924 | 0.905 | 0.898 | **0.775** | **0.787** | **0.775** | **0.871** |
| | ✓ | ✓ | ✓ | 0.926 | **0.905** | **0.901** | 0.775 | 0.783 | 0.768 | 0.868 |

Table 18: The performance in AUROC for our different baseline modeling strategies, split by PE status. **CT** is the CTPA based LRCN model, **M** is the structured EHR based MOTOR model, and **G** is the structured EHR based gradient-boosted trees model. The best overall models are **bolded** and the best individual models are underlined.

| # | PESI Score criteria |
|---|---|
| 1 | Age > 80 |
| 2 | History of Cancer |
| 3 | History of Chronic Cardiopulmonary Disease |
| 4 | Heart Rate (bpm) $\geq$ 110 |
| 5 | Systolic BP (mmHg) < 100 |
| 6 | $O_2$ Saturation < 90% |

Table 19: Criteria for PESI score

| sPESI Score | # Cases With Score |
|---|---|
| 0 | 169 |
| 1 | 361 |
| 2 | 529 |
| 3 | 450 |
| 4 | 184 |
| 5 | 30 |
| 6 | 1 |

Table 20: The statistics for the sPESI scores on the 1,719 cases in our cohort that it can be calculated on.

| Input Modality | | | | Prognostic | | | | | | |
|---|---|---|---|---|---|---|---|---|---|---|
| Image | EHR | | | In-Hospital Mortality | | | Re-admission | | | PH |
| CT | M | G | P | 1 m | 6 m | 12 m | 1 m | 6 m | 12 m | 12 m |
| ✓ | | | | 0.676 | 0.634 | 0.663 | 0.643 | 0.478 | 0.631 | 0.579 |
| | ✓ | | | **0.808** | **0.813** | **0.787** | **0.745** | **0.684** | **0.678** | **0.725** |
| | | ✓ | | 0.749 | 0.749 | 0.741 | 0.679 | 0.620 | 0.619 | 0.688 |
| | | | ✓ | 0.569 | 0.571 | 0.571 | 0.701 | 0.679 | 0.614 | 0.567 |

Table 21: The performance in AUROC for our different baseline modeling strategies given patients with PE. Note that this set of evaluations is only done on the subset of cases that have both PE and enough data for the simplified PESI risk score. **CT** is the CTPA based LRCN model, **M** is the structured EHR based MOTOR model, **G** is the structured EHR based gradient-boosted trees model, and **P** is the simplified PESI risk score. The best models are **bolded**.

than the prospective studies used to generate and validate sPESI. For example, our ability to extract features like the history of Chronic Cardiopulmonary Disease is relatively limited as we are restricted to structured data already within the health record.

# K  Additional Metrics

For our main analysis, we compare models in terms of AUROC as it is a low variance and widely used metric. However, additional metrics, especially in the clinical space, can also be important when evaluating the utility of models.

| Input Modality | | | Diagnostic | Prognostic | | | | | | |
|---|---|---|---|---|---|---|---|---|---|---|
| Image | EHR | | PE | In-Hospital Mortality | | | Re-admission | | | PH |
| CT | M | G | (+) | 1 m | 6 m | 12 m | 1 m | 6 m | 12 m | 12 m |
| ✓ | | | 0.463 | 0.189 | 0.288 | 0.324 | 0.056 | 0.124 | 0.169 | 0.230 |
| | ✓ | | 0.327 | 0.396 | 0.537 | 0.588 | 0.160 | 0.342 | 0.402 | 0.485 |
| | | ✓ | 0.335 | 0.234 | 0.426 | 0.497 | 0.145 | 0.276 | 0.334 | 0.582 |
| ✓ | ✓ | | 0.510 | 0.426 | **0.545** | **0.599** | 0.164 | 0.337 | 0.393 | 0.481 |
| ✓ | | ✓ | 0.515 | 0.295 | 0.447 | 0.521 | 0.145 | 0.271 | 0.330 | 0.573 |
| | ✓ | ✓ | 0.354 | 0.399 | 0.542 | 0.587 | 0.179 | **0.346** | **0.407** | **0.597** |
| ✓ | ✓ | ✓ | **0.523** | **0.428** | 0.542 | 0.598 | **0.180** | 0.343 | 0.399 | 0.589 |

Table 22: The performance in AUPRC for our different baseline modeling strategies, including late fusion. **CT** is the CTPA based LRCN model, **M** is the structured EHR based MOTOR model, and **G** is the structured EHR based gradient-boosted trees model. The best overall models are **bolded** and the best individual models are underlined.

We thus perform additional analysis to compare our models in terms of both AUPRC (area under the precision-recall curve) (see Table 22 and ECE (expected calibration error) (see Table 23). AUPRC provides an estimate of the precision of a model under various recall thresholds and ECE provides an estimate of the calibration of a model. We use 10 bins for our ECE estimate.

The relative model performance rankings for both of these additional metrics are very similar to the rankings seen with AUROC, with the EHR models doing better at prognostic tasks while the image models do better at the diagnostic task.

| Input Modality | | | Diagnostic | Prognostic | | | | | | |
|:---:|:---:|:---:|:---:|:---:|:---:|:---:|:---:|:---:|:---:|:---:|
| **Image** | **EHR** | | **PE** | **In-Hospital Mortality** | | | **Re-admission** | | | **PH** |
| CT | M | G | (+) | 1 m | 6 m | 12 m | 1 m | 6 m | 12 m | 12 m |
| ✓ | | | 0.278 | 0.423 | 0.369 | 0.265 | 0.136 | 0.347 | 0.346 | 0.325 |
| | ✓ | | 0.026 | 0.011 | 0.010 | 0.015 | 0.008 | **0.004** | **0.007** | 0.012 |
| | | ✓ | 0.015 | 0.020 | 0.053 | 0.017 | 0.016 | 0.016 | 0.026 | 0.021 |
| ✓ | ✓ | | 0.024 | **0.004** | 0.012 | 0.017 | 0.009 | 0.009 | 0.010 | 0.016 |
| ✓ | | ✓ | 0.015 | 0.007 | 0.019 | 0.017 | **0.004** | 0.009 | 0.014 | 0.023 |
| | ✓ | ✓ | **0.007** | 0.004 | **0.010** | **0.014** | 0.006 | 0.009 | 0.014 | **0.009** |
| ✓ | ✓ | ✓ | 0.016 | 0.005 | 0.011 | 0.015 | 0.006 | 0.009 | 0.013 | 0.025 |

Table 23: The calibration performance in ECE for our different baseline modeling strategies, including late fusion. Lower scores indicate better models with this metric. **CT** is the CTPA based LRCN model, **M** is the structured EHR based MOTOR model, and **G** is the structured EHR based gradient-boosted trees model. The best overall models are **bolded** and the best individual models are underlined.

