# OpenReview forum: "INSPECT: A Multimodal Dataset for Pulmonary Embolism Diagnosis and Prognosis"
_NeurIPS.cc/2023/Track/Datasets_and_Benchmarks — NeurIPS 2023 Datasets and Benchmarks Poster_

### Official Review · Reviewer_PLZb · 2023-07-13
**A potentially high-impact dataset and benchmark if remaining issues are fixed**

**Rating:** 7
**Confidence:** 4
**Clarity:** The main paper is very well-written. …

**Strengths:**

- Uniqueness and size of the dataset: The dataset is, to my knowledge, the largest of its kind.
- Uncertainty quantification: Bootstrapping on the test set is performed to quantify validation uncertainty.
- Open science: To increase reproducibility, the authors also release the trained model weights
- Interesting results: The results on the multi-modal approach are interesting and could trigger further research
- Data sheet: Availability of a data sheet

**Additional Feedback:**

I am in strong favor of the dataset to be released eventually although I do not think it's ready yet (see above). I will carefully read the rebuttal and am ready to move my score up if the issues can be resolved.

**Correctness:**

I have already mentioned most concerns above. On top of that: The authors state that RSNA RESPECT (used for finetuning the sequence encoder task) and RadFusion are parcels of INSPECT and that they garantee a clear separation between training and testing data. In table 1 they show the demographics for these overlapped studies. But the number of cases of RadFusion (1011) are not the ones reported in the original study (1794).The methods are mostly sound but several issues should be fixed as details above.

**Documentation:**

- Some issues with access to the data, licencing, and maintenance are mentioned in “Opportunity for Improvements”
- The authors include a Licence Terms of Use detailing intended use of the dataset, as well as an interactive notebook detailing the structure of the data.
- For benchmarks, code was provided for reproducibility, however, without having access to the data I was not able to run it to verify.

There seem to be several inaccuracies in the datasheet:
- While the datasheet states “The code for data collection can be found here:” the link seems to be missing.
- The datasheet asks for exact wording of notification about data collection, but the authors only mention a signed privacy notice, not providing wording. Similarly for individual’s consent for data collection. They instead refer to their approval by their university’s administrative panel on human subject research.
- The authors provide details of people responsible for the maintenance of the dataset, but do not provide details on how often, by whom, or how the changes would be communicated, only stating that it would be updated “on a regular release basis”.  They inform that change logs will be provided, should any changes to the dataset occur, but do not inform where to find them.

MINOR: The authors state “As mentioned above, we extract the data using STARR”, however there seems not to be such mention earlier in the datasheet (there is in the main text and supplement).

**Ethics:**

-

**Limitations:**

Limitations have partially been addressed (not the ones mentioned under “Opportunities for Improvement”)

**Opportunities For Improvement:**

- Data access: I could not find a representative part of the dataset online. On  https://stanfordaimi.azurewebsites.net, I found RadFusion, which, however, does not include snippets from radiology reports.
- Lack of DOI: The dataset does not have a doi. How is permanent accessibility ensured exactly? I would be very much in favor of adhering to the data hosting standards put forth by Nature Scientific Data (https://www.nature.com/sdata/policies/repositories), specifically (1) Ensure long-term persistence and preservation of datasets in their published form (2) Provide stable persistent identifiers for submitted datasets (e.g. Datacite DOIs) (3) Allow public access to data without barriers, such as logins or paywalls.

- License: What was the rationale for not choosing a more flexible license, e.g. CC0 or CC-BY or at least CC BY-NC-SA  or CC BY-NC-ND? A standard license could also help resolve ambiguities. For example, the authors write “you must not modify, reverse engineer, decompile, or create derivative works from the INSPECT dataset”. This might be interpreted as interfering with data augmentation.
- Dataset update/reproducibility: The authors write “The dataset will be updated on the AIMI website as needed, on a regular release basis, depending on dataset errors or other corrections identified by users of the dataset.” Given that the dataset does not have a DOI, how will the version management be performed? For example, if somebody published a paper based on a previous version, how can reproducibility be guaranteed?
- Metrics: Could the authors comment on their choice of metrics? Specifically
An increasing body of work shows that metrics are often not well-aligned with the clinical interest (see e.g. the Metrics Reloaded project: https://arxiv.org/abs/2206.01653). What was the authors’ procedure for choosing metrics? Why was no emphasis put on the calibration of the algorithms (as recommended as one important aspect in Metrics Reloaded, for example)?
The metrics used for the validation of automatic labels (Appendix) differed from task to task? Why? What was the rationale for focussing on AUROC in the first experiment and not reporting it at all in the second one?
- Uncertainty of labels: Some of the labels of the dataset were generated with an ML algorithm to overcome the need for manual labeling. While the authors do provide an experiment to estimate the performance of the method, I would find it even more convincing to provide expert labels for the test set.
- Ambiguous labels: In cases of ambiguity, the authors went for a negative label -> why not use “unsure” instead? I am referring to sentences like “ "Negative" is that the patient either doesn’t have pulmonary hypertension or it is unknown”
- Ontologies: The ontologies used to describe the data are not detailed in any of the documents. The most descriptive sentence I found was “Medical codes are symbols drawn from controlled vocabularies (e.g., SNOMED, ICD, LOINC)“ plus Tab.9 in the appendix . Rather than putting “e.g.”, the authors are invited to specify the exact format and all ontologies applied.

MINOR

- “We defined splits based on patient ids, such that each patient only appears in one split.” -> How come that the the prevalences in the validation and test split (Tab. 2) are identical? Apparently, prevalences were also used for defining the splits? This has also not been detailed in the datasheet although it is very important.
- there are several language (e.g. inconsistent tenses) and formatting issues in the datasheet document.
- I was not sure whether the authors sometimes used the term “anonymized” when they meant “pseudomized?
- “as a function of the test set” confused me in Tab. 12. It should be as a function of the model, no?
- The authors may want to reconsider the tenses used in the manuscript (past tense for data acquisition, processing etc)
- The acronym DICOM is used used multiple times in section 3 before being defined in section 4
- Some of the used benchmark architectures: Clinical Longformer and ResNetV2 lack references.

Li, Yikuan et al. “Clinical-Longformer and Clinical-BigBird: Transformers for long clinical sequences.” ArXiv abs/2201.11838 (2022)

He, Kaiming et al. “Identity Mappings in Deep Residual Networks.” European Conference on Computer Vision (2016).

**Relation To Prior Work:**

Yes, the relation to prior work, especially to the UK Biobank, is well-explained.

**Summary And Contributions:**

The contribution of the paper is two-fold. (1) The authors present the benchmarking dataset INSPECT, containing computed tomography (CT) images, sections of radiology reports and electronic health record (EHR) data and of more than 20,000 pulmonary embolism (PE) patients. (2) Based on this multimodal dataset they present a benchmark comprising eight diagnostic and prognostic tasks relevant in the context of PE and provide reference results for several image only, EHR only and multimodal baseline models. The results suggest that the integration of medical imaging and structured EHRs improves performance in diagnosing PE and that further work is necessary to leverage a multimodal approach for prognostic tasks.

---

> ### Author Response · Authors · 2023-08-21
>
> # Response to Reviewer PLZb
>
> Dear Reviewer PLZb,
>
> Thank you for your review. We address each of your points below:
>
> ### Representative subset data access:
>
> We apologize for the confusion and have released a sample dataset per the main rebuttal.
>
> ### Dataset DOI:
>
> We agree with the reviewer and will work with the AIMI center to assign a DOI.
>
> ### Dataset license:
>
> Thank you for these comments. To protect patient privacy, our licensing must restrict redistribution of the INSPECT dataset. This is unfortunately at odds with Creative Commons licensing, which allows redistribution. INSPECT will be released under a license that follows the best practices established by PhysioNet and their PhysioNet Restricted Health Data License 1.5.0. We are in the process of finalizing a Stanford version of this license.
>
> The licensing will not prohibit use that is central to training ML models, e.g., data augmentation.
>
> ### Dataset update / reproducibility:
>
> As a condition of release, per the university and the AIMI center, we have a protocol for dataset maintenance based on discovery of patient health information in the released dataset. Per licensing terms, users must notify the AIMI center of any protected health information found in the dataset. This will trigger our team and the AIMI center updating the dataset to remove said PHI information and a request to users who have signed the DUA to delete specific records. Dataset versions and a changelog will be maintained as part of the dataset release itself. We note this is standard practice for healthcare data releases. We anticipate the dataset will remain largely static, based on experiences with current AIMI hosted datasets such as CheXpert.
>
> ### Choice of metrics:
>
> We agree that AUROC is not sufficient for all use cases so we have added AUPRC (area under the precision recall curve) and ECE (expected calibration error) to our benchmark, with tables for those metrics included in the updated appendix section K. For the NLP labeler, we use metrics (i.e. F1, precision, recall, accuracy) to facilitate comparison to prior NLP work.
>
> ### Uncertainty of labels
>
> Thank you for the suggestion. While we generally agree that expert-annotated test sets should be preferred, we found this underlying NLP task to be quite simple [1] even for older generation ML models and more so for our more powerful Clinical Longformer model. We did an error analysis in Supplementary section D1 and found our model performs near perfectly. Given this, we felt it wasn't the best use of expert (and expensive) radiologist annotation effort.
>
> Additionally, we made sure to exclude RSNA and Radfusion test/valid set samples from our training set, so users can test their model’s PE performance on these datasets.
>
> [1] Banerjee, Imon, et al. "Comparative effectiveness of convolutional neural network (CNN) and recurrent neural network (RNN) architectures for radiology text report classification." Artificial intelligence in medicine 97 (2019): 79-88.
>
> ### Ambiguous labels
>
> Thank you for this suggestion. We originally considered a more nuanced labeling schema for pulmonary hypertension, partially outlined in Appendix D.2. This would include labels such as unknown or uncertain. However, due to the time required to manually label and verify a PH NLP labeler, we only constructed a test set for validating the quality of ICD-based PH labels. As future work we would like to use a combination of NLP and medical codes to derive more nuanced labels.
>
> ### Ontologies
>
> In total we used the 18 ontologies from OHDSI’s Athena ontology collection (athena.ohdsi.org) including publicly specified ontologies (such as LOINC / SNOMED) and some are OHDSI specific (such as Ethnicity, Race, and Visit). The full list is in Table 5 in supplementary section A.
>
> ### Validation and test split prevalence
>
> We want to emphasize that we only create splits based on patients, and that the prevalence is not identical in Table 2. We adhere to the canonical splits in terms of patients to avoid data leakage.
>
> ### Inconsistent tenses
>
> We have made changes to the original manuscript to fix several inconsistent tenses and formatting issues.
>
> ### Anonymized vs “pseudomized”
>
> Thank you for this clarification. Our dataset is de-identified in compliance with HIPAA standards (i.e., the Safe Harbor method), the standard for de-identification in the United States.  We have updated the manuscript to be more clear in our word choice to indicate when we mean this type of pseudomization.
>
> ### Function of the test set vs function of the model .
>
> We apologize for the unclear language. Our confidence estimate using a test set bootstrap of 1000 samples. Our EHR baselines, LightGBM and linear probing based on a fixed MOTOR model, are deterministic so have no variance with respect to any random seed. We have however added a new analysis to look at the standard deviation of our image models with respect to the random seed used to train them as Table 16 in the appendix.

---

> > ### Author Response · Authors · 2023-08-21
> >
> > # Response to Reviewer PLZb - Part 2
> >
> > ### Defining acronym DICOM
> >
> > The manuscript is now updated to define DICOM in section 3.
> >
> > ### Benchmark architecture references
> >
> > The references for these works are now added.
> >
> > ### Inaccuracies in the datasheet
> > We apologize for this oversight. The full patient notice of privacy form is at https://stanfordhealthcare.org/content/dam/SHC/patientsandvisitors/patient-privacy/docs/100-598-noticeofprivacypractices-english.pdf
> >
> > The relevant section is: “As part of an academic medical center, the Hospital has an active research program. For example, research is ongoing to advance care, to evaluate investigational procedures to treat conditions, to compare the health of patients who have received one medication with those who have received another medication for the same condition, and to learn from medical record studies. We generally ask for your written authorization before using your health information or sharing it with others to conduct research. Under limited circumstances, we may use and disclose your health information without your authorization. In most of these latter situations, we must comply with law and obtain approval through an independent review process to ensure that research conducted without your authorization poses minimal risk to your privacy.”
> >
> > Our use is one such limited circumstance and has been approved by our IRB.

---

> > ### Comment · Reviewer_PLZb · 2023-08-24
> > **Majority of comments addressed**
> >
> > The authors have addressed the majority of my comments in their revision while my main concern (no access to complete data set, no doi) could not yet be resolved. However, given the uniqueness of the data and the challenges involved in releasing medical data, I will move my score up in good faith.

---

### Official Review · Reviewer_CGfA · 2023-07-20
**Large Dataset with Data Access Concern**

**Rating:** 3
**Confidence:** 5
**Correctness:** Yes. The curation and data cleaning p…
**Clarity:** The paper is well written.

**Strengths:**

A large multimodal dataset with 20,028 patients for pulmonary embolism (PE) including CT pulmonary angiogram (CTPA), radiology report, and electronic health record (EHR) data.

The dataset curation and cleaning process is well-justified and clearly documented.

Comprehensively benchmarking the dataset.

**Additional Feedback:**

None.

**Documentation:**

There is no data link. The authors should make all data available to reviewers (not just a subset) by the end of their rebuttal period. Otherwise, they may give reviewers the impression that the dataset is not available.

**Ethics:**

Not really. But the authors should explain how this dataset is different from the RSNA PE dataset (See more details in Opportunities For Improvement).

**Limitations:**

The limitation of this dataset is discussed well. The authors discussed potential label errors produced by NLP and the limitation of data from a single site (Stanford Medicine).

**Opportunities For Improvement:**

The authors should show some CTPA images in the paper as most people have no clue about what CTPA looks like.

The authors should have made all data available to reviewers at the time of submission. Yet, they did not. To make up for this mistake, the authors should make all data available to reviewers (not just a subset) by the end of their rebuttal period. Otherwise, they may give reviewers the impression that the dataset is not available.

The authors should make it crystally clear how this dataset is different from the RSNA PE dataset (https://pubs.rsna.org/doi/full/10.1148/ryai.2021200254). This is also a major concern.

In Table 3, it seems the imaging data does not provide additional value for predicting outcomes. It compromises the value of this dataset as the main claim is that this dataset includes 3D CTPA data.

The authors should provide a table to document prior large multimodal medical imaging datasets and highlight how this dataset is different and valuable in comparison.

**Relation To Prior Work:**

The authors should provide a table to document prior large multimodal medical imaging datasets and highlight about how this dataset is different and valuable in comparison.

**Summary And Contributions:**

The authors present a large multimodal dataset with 20,028 patients for pulmonary embolism (PE) including CT pulmonary angiogram (CTPA), radiology report, and electronic health record (EHR) data. The main contribution of this paper is the large multimodal dataset for PE. But on the other hand, I have concerns about data access. The authors say "We will work to provide a subset to reviewers", but there is no link to the subset. As a dataset paper, this is a major issue for not having data available to reviewers. Another issue that is even more severe is that in the RSNA study https://pubs.rsna.org/doi/full/10.1148/ryai.2021200254, the authors already released PE data from 12 ,195 patients. The authors should make it very clear how this dataset is different from the RSNA PE dataset. If the main difference is just the data size increases from 12 ,195 to  20,028 patients, then the originality and novelty of this paper are dampened.

---

> ### Author Response · Authors · 2023-08-21
>
> # Response to Reviewer CGfA
>
> Dear Reviewer CGfA
>
> Thank you for your thoughtful feedback on our paper. We provide point-by-point responses to your comments below.
>
> ### Showing example CTPA images
>
> We agree with the reviewer that some readers might not be familiar with CTPA. We now include a sample image and description in our manuscript, detailed in supplementary section E.
>
> ### Data availability
>
> We agree that access to a representative sample of the dataset is critical for the reviewing process and sincerely apologize about delays in providing said sample to reviewers. Per our overall rebuttal, we have included a link to a sample dataset for review and provided an updated timeline for the dataset release.
>
> ### Difference between INSPECT and RSNA PE
>
> We thank the reviewer for pointing out this ambiguity. There are several key differences between the RSNA PE dataset and INSPECT:
> - RSNA is a multi-institution dataset (5 institutions) containing 12,195 imaging studies. INSPECT is a single institution dataset containing 23,804 studies. Only 601 cases (imaging studies) are shared across datasets (see Table 2). No RSNA data is included in the INSPECT validation or test sets and no RSNA test set data is included in INSPECT.
> - RSNA includes a single case per patient. INSPECT can include multiple cases per patient.
> - RSNA provides a single modality (CT scans). INSPECT is multimodal, including CT scans, paired radiology note impressions (i.e., unstructured text describing imaging findings), and longitudinal EHR records (i.e., all pre and post scan medical record data which includes diagnoses, medications, vitals, lab test values, demographics, etc.)
> - RSNA includes expert-annotated diagnostic image labels, i.e., labels for PE-related pathologies discernible in pixel data.  INSPECT includes both (1) diagnostic labels extracted from radiology reports (with verified NLP performance of extracted labels) and (2) prognostic labels for future outcomes.
> We have updated the manuscript to make sure these differences are clearly communicated and indicated in our dataset which samples are in RSNA PE.
>
> ### The benefit of image modality
>
> We appreciate the insightful comment on the relative benefits of incorporating imaging vs. other modalities into our models. For many medical diagnosis and prognosis tasks, synthesizing information from both medical images and patient EHR is crucial for both physicians and AI models. For PE tasks, we discuss current medical knowledge that suggests possible benefits in incorporating image data (line 76). However, we emphasize that the current work only includes simple baseline imaging and late fusion methods. A primary motivation of INSPECT is to encourage researchers to develop new computational methods for multimodal fusion that better capture synergies between modalities. In addition, by providing the entire patient timeline, researchers can also use our dataset to curate other outcome labels that require using both image and EHR data.
>
> ### Comparisons with other multimodal healthcare datasets
>
> We are grateful to the reviewer for providing this valuable suggestion. In response, we have incorporated a comparison to prior work into our revised manuscript and codebase (https://github.com/som-shahlab/inspect). Please also find the table for the general response to all reviewers.

---

### Official Review · Reviewer_vbzo · 2023-07-21
**A multimodal dataset for Patient Outcome Prediction of Pulmonary Embolisms**

**Rating:** 7
**Confidence:** 4

**Strengths:**

- Largest multimodality dataset that combines the EHR data and 2d/3d imaging data together as far as I know.
- Clear and curated code and documents, one of the most transparent research.
- Well-written, clear, and easy to understand. I appreciate the *Discussion* section with almost everything considered.

**Additional Feedback:**

Is it possible to construct the labels with uncertainty measures?

**Clarity:**

The paper is well-written, clear, and easy to understand. The authors have provided sufficient detail about the dataset, the methodology used to create it, and the benchmarking process. The code is also written clearly with nice comments.

**Correctness:**

The claims made in the submission appear to be correct. The dataset is constructed in a sound manner, adhering to ethical and privacy considerations. The evaluation methods and experiment design for the benchmark are also appropriate and performed correctly.

**Documentation:**

Yes

**Ethics:**

The authors have addressed ethical concerns related to data privacy and patient consent. They have also specified that the dataset is intended for research use and not for direct clinical decision-making.

**Limitations:**

The authors have acknowledged the limitations of their work and potential negative societal impacts. They also address potential limitations related to the single-site nature of the data, the potential for errors in label construction, and the absence of full radiology reports due to de-identification protocols.

**Opportunities For Improvement:**

- The error bars for each prediction performance (mean and std) is not reported. K (e.g. K=10) different seeds for model initialization is essential to evaluate the stability of the baseline models.
- Lack more statistical comparisons with other multimodality healthcare datasets and benchmarks. For example the *Comparison to Prior Work* table in [Link](https://github.com/som-shahlab/ehrshot-benchmark)
- The late fusion module can be more clarified. How to take a weighted mean? (Line 229, 4.4.3 Model Fusion)
- As the authors discussed, the generated NLP labels may introduce bias or be miscalculated. It will be helpful to add an illustration figure of the labeling process, and examples of discovered wrong labels at the initial stage. (Authors detailedly discussed in appendix D, trying hard to mitigate it, 👍)

**Relation To Prior Work:**

The authors have clearly distinguished their work from previous contributions. They have provided a comprehensive review of existing datasets and addressed the gaps in those datasets that their work aims to fill.

**Summary And Contributions:**

The paper introduces a large-scale multimodal dataset named INSPECT from a cohort of pulmonary embolism (PE) patients. The dataset has ~20K patients including CT images (2D/3D), sections of radiology reports and structured EHR data (including demographics, diagnoses, procedures, and vitals). It is the largest multimodal dataset for integrating 3D medical imaging and EHR data. The authors also provides a comprehensive benchmark for PE diagnosis and prognosis with easy-to-reproduce code and implementation details. From the manuscript and submitted supplementary material, this work is very transparant and friendly to healthcare AI researchers.

---

> ### Author Response · Authors · 2023-08-21
>
> # Response to Reviewer vbzo
>
> Dear Reviewer vbzo
>
> We appreciate your insightful comments and the time you invested in reviewing our manuscript. Below, we address each of your points of concern.
>
> ### Missing error bars with respect to the training seed
>
> We agree that adding error bars with respect to the randomness inherent in training is essential to evaluate the stability of comparisons done in our benchmark. We ran 10 replicates of our CT/imaging baselines and have included the corresponding numbers within Table 16 in the appendix. Our EHR baselines, LightGBM and linear probe models based on MOTOR, are deterministic, so have no random seed parameter. (MOTOR itself is not deterministic in that retraining the foundation model can result in different numbers, but results given a fixed foundation model are deterministic.)
>
> ### Lacking comparisons with other multimodal healthcare datasets**
>
> We thank the reviewer for this valuable suggestion. We have created a comparison to prior work and added it to our updated manuscript and codebase. Please also find the table to the general response to all reviewers.
>
> ### Clarification on late fusion modules.
>
> We apologize for the ambiguity. Our late fusion model is a weighted ensemble model, where weights for each modality’s outputs are learned on our validation set using logistic regression.  We have updated our manuscript to clarify our strategy for late fusion.
>
> ### Illustration figure of the labeling process.
>
> We agree with the reviewer that an illustration of our labeling process could be helpful and have updated our manuscript to include Figure 6 in the Appendix outlining our labeling process. We have added some discussion of error modes of our NLP labeler in section D.1 and Figure 7.

---

### Author Response · Authors · 2023-08-21
**General Response to All Reviewers**

# General Response to All Reviewers:

Dear reviewing committee,

We thank the reviewers for their thoughtful feedback and constructive criticisms of our manuscript. We appreciate all reviewers for acknowledging the uniqueness and size of our dataset, which we believe is a significant contribution to the field. INSPECT is, to our knowledge, the largest multimodal 3D medical dataset available to researchers. We hope that by releasing radiology text, longitudinal EHRs, and CT scans, we can facilitate new methodological innovations in medical multimodal learning.

We have synthesized common feedback from reviewers as point-by-point responses below, to provide clarity on our methodology and the significance of our contribution.

## Dataset Release

We acknowledge that access to a representative sample of the dataset is critical for the review process and sincerely apologize for the delays in providing a sample to reviewers. We encountered some unanticipated delays during our institutional review process, specifically due to the breadth of longitudinal EHR data included in INSPECT. However, we have formal approval from our OGC/UPO (Office of General Counsel, and University Privacy Office), the University Research IT department,  and the hosting site AIMI center to release this dataset. We are working with all teams to ensure that the entire dataset will be ready for release as quickly as possible. For reviewer purposes:

- We now have a sample of 122 patients from INSPECT for feedback. This includes CT scans, radiology impression sections, demographics, and diagnostic/prognostic labels. We also provide a single EHR sample to illustrate the form the final dataset release will take. The dataset is available at https://stanfordaimi.azurewebsites.net/datasets/521d0d11-7c3c-45ef-8b99-a49cd70cba2c
- Our updated timeline for release
    - October 1st: Finalize dataset license approval
    - November 1st: Complete full dataset review
    - December 1st: Release full dataset before the conference
- Should this timeline change for any reason, we will communicate the updated release schedule to NeurIPS organizers.

We appreciate reviewers’ patience as we work to ensure that the dataset is released in a manner that protects patient privacy and rights while also fulfilling the needs of the research community. We hope that this sample dataset allows reviewers to assess the unique advantages INSPECT brings to multimodal machine learning in medicine.

## Clarifying INSPECT’s Contributions vs. Existing Multimodal Datasets
We have added a table summarizing existing multimodal medical datasets to emphasize INSPECT’s unique contributions.

|Dataset|Imaging Modalities|Reports|EHR|#Patients|#Image Studies|Diagnostic Tasks|Prognostic Tasks|
|-|-|-|-|-|-|-|-|
|Open-I|Chest X-ray|Yes|-||7,466|-|-|
|CheXpert|Chest X-ray|-|-|65,240|224,316|14|-|
|MIMIC-CXR|Chest X-ray|Yes|Yes|65,379|227,835|14|-|
|UK Biobank Imaging|Multiple MRI, DXA, Ultrasound|-|Partial|100,000|Many|*|*|
|RSNA PE|CT|-|-|12,195|12,195|13|-|
|RadFusion|CT|-|Partial|1,794|1,837|1|-|
|INSPECT (Ours)|CT|Yes|Yes|20,028|23,804|2|3|

## RSNA PE vs INSPECT
Reviewers expressed concern about the novelty of INSPECT vs. RNSA PE (a prior dataset release of CT scans for PE patients). We have clarified the differences between the datasets in the manuscript, primarily that INSPECT is a single institution dataset that includes 3 modalities (EHR, text, CT scans) while RSNA is a multi-institutional dataset containing only CT scans. We also clarify that INSPECT is almost 2x larger than RSNA while only overlapping RSNA by 601 cases.

## Replicates of ML Baselines
While we initially provided bootstrap confidence intervals (see Tables 14-15) for all results, we did not run model replicates to assess variance due to the choice of random seed. For our EHR results, both of our baselines, LightGBM and linear probe models based on MOTOR, are deterministic. (MOTOR itself is not deterministic in that retraining the foundation model can result in different numbers, but results given a fixed foundation model are deterministic.) For imaging results, we agree replicates would improve the rigor of our baselines, and we have done that analysis using 10 random seeds. Table 16 in the appendix contains the mean performance and standard deviation as a function of the random seeds for the image models.

---

> ### Comment · Reviewer_CGfA · 2023-08-29
> **Still Have Dataset Access Concerns and the Dataset's Similarity with RSNA Dataset**
>
> I gave a score of "5: Marginally below acceptance threshold" with an anticipation that the authors would completely address the issue of data availability. Unfortunately, the data access issue is still not really addressed:
>
> 1. This is the NIPS Datasets track, therefore having the complete dataset available to the reviewers by the time of the final decision is essential, as the paper's value is all about the dataset. Having a timeline for data release after the final decision is not helpful as there is no way reviewers can really hold the authors accountable in case they are not able to follow their proposed timeline for data release due to various reasons such as "We encountered some unanticipated delays during our institutional review process". Again, in my view, for all papers submitted to the NIPS Datasets track, the complete datasets should be available to reviewers by the time of final decision.
>
> 2. Downloading data "https://stanfordaimi.azurewebsites.net/datasets" requires the reviewers to fill in personal information including name, email, phone number, and institution information, otherwise the data cannot be downloaded. This is a clear breach of the anonymous review rule. This should never happen for a NIPS Dataset paper.
>
> 3. The authors' clarifications about the difference between the INSPECT and RSNA datasets make it abundantly clear: the RSNA dataset is actually very similar to the INSPECT dataset proposed in this paper, except the RSNA dataset contains fewer subjects and fewer diagnostic attributes with only cross-sectional data. I understand that the RSNA dataset is multi-institutional and the INSPECT dataset is from a single institution, and only 601 cases overlap between the two datasets. However, the difference is rather minimal, and the novelty of this dataset is therefore compromised.
>
> Based on the justifications above, I will lower my score to rejection.

---

> > ### Author Response · Authors · 2023-08-29
> >
> > We thank the reviewer for their comments and have addressed each below (we also highlight our original, more detailed response to the reviewer's original review in the thread below). We hope that the reviewer considers our points in their overall assessment and score.
> >
> > > This is the NIPS Datasets track, therefore having the complete dataset available to the reviewers by the time of the final decision is essential
> >
> > We appreciate this position. We have endeavored to make a representative sample of the dataset available for reviewers and outlined a timeline for release of the entire dataset. We hope that the reviewer appreciates the unique difficulties present in releasing healthcare datasets for machine learning research, which requires navigating substantial governance considerations to ensure patient privacy. This is even more challenging when releasing multimodal data such as text and structured EHRs. These barriers are in part why there are so few multimodal healthcare datasets available to researchers.
> >
> > > Downloading data "https://stanfordaimi.azurewebsites.net/datasets" requires the reviewers to fill in personal information including name, email, phone number, and institution information
> >
> > We apologize for the appearance of impropriety here and not communicating the access protocol clearly. We emphasize that we, as the authors of this manuscript, do not manage or have access to the AIMI center's hosting platform. This is managed by external parties who conduct dataset review for PHI, etc and manage file uploading and access. As such, we never see the users who apply for access to any dataset. This preserves the anonymous review rule. To protect patient privacy, AIMI requires the ability to confirm the identity of the person downloading the data. This is a strict requirement.
> >
> > > The authors' clarifications about the difference between the INSPECT and RSNA datasets make it abundantly clear: the RSNA dataset is actually very similar to the INSPECT dataset proposed
> >
> > We emphasize again that INSPECT is multimodal while RSNA PE only contains imaging. INSPECT contains unstructured text (radiology impression sections) and longitudinal structured EHR data for 20k patients. This makes INSPECT more like MIMIC-III in scale (which has 34k patients). This is a significant difference. Imaging-only datasets such as as RNSA PE limit the types of research questions that can be explored and our hope is that INSPECT opens up new opportunities to study multimodal fusion. For example, INSPECT enables exploring 3D contrastive learning between report text and CT scans (currently not possible given research datasets). INSPECT further enables exploring downstream forecasting of future health events based on imaging data, given our longitudinal EHR data. This is not possible given an image only dataset that reflect a single event/point in a patient timeline.

---

### Decision · Program_Chairs · 2023-09-22

**Decision:**

Accept (Poster)

**Comment:**

The paper introduces a large-scale multimodal dataset named INSPECT from a cohort of pulmonary embolism (PE) patients.
The dataset has information from 20000 patients. It includes CT images (2D/3D), sections of radiology reports and structured EHR data (including demographics, diagnoses, procedures, and vitals). It is the largest multimodal dataset for integrating 3D medical imaging and EHR data.
The dataset is interesting and relevant.
Nevertheless, there are issues related to the timeline to make the data available.